# LECT2 drives haematopoietic stem cell expansion and mobilization via regulating the macrophages and osteolineage cells

Xin-Jiang Lu[1], Qiang Chen[1], Ye-Jing Rong[1], Guan-Jun Yang[1], Chang-Hong Li[1], Ning-Yi Xu[2], Chao-Hui Yu[3], Hui-Ying Wang[4], Shun Zhang[5], Yu-Hong Shi[1] & Jiong Chen[1]

Haematopoietic stem cells (HSCs) can differentiate into cells of all lineages in the blood. However, the mechanisms by which cytokines in the blood affect HSC homeostasis remain largely unknown. Here we show that leukocyte cell-derived chemotaxin 2 (LECT2), a multifunctional cytokine, induces HSC expansion and mobilization. Recombinant LECT2 administration results in HSC expansion in the bone marrow and mobilization to the blood via CD209a. The effect of LECT2 on HSCs is reduced after specific depletion of macrophages or reduction of osteolineage cells. LECT2 treatment reduces the tumour necrosis factor (TNF) expression in macrophages and osteolineage cells. In TNF knockout mice, the effect of LECT2 on HSCs is reduced. Moreover, LECT2 induces HSC mobilization in irradiated mice, while granulocyte colony-stimulating factor does not. Our results illustrate that LECT2 is an extramedullar cytokine that contributes to HSC homeostasis and may be useful to induce HSC mobilization.

[1] Laboratory of Biochemistry and Molecular Biology, Department of Marine and Life Sciences, Ningbo University, Ningbo 315211, China. [2] Life Sciences Institute, Zhejiang University, Hangzhou, Zhejiang 310058, China. [3] Department of Gastroenterology, the First Affiliated Hospital, School of Medicine, Zhejiang University, Hangzhou 310003, China. [4] Department of Allergy and Clinical Immunology, the Second Affiliated Hospital, School of Medicine, Zhejiang University, Hangzhou 310003, China. [5] Clinical Research Center, Ningbo No. 2 Hospital, Ningbo 315010, China. Correspondence and requests for materials should be addressed to J.C. (email: jchen1975@163.com).

Haematopoietic stem cells (HSCs) are used in clinical transplantation protocols for the treatment of a wide variety of immune-related diseases[1,2]. The initial source of HSCs is the bone marrow (BM), but HSCs can also be obtained from the peripheral blood, following mobilization procedures[2]. HSC expansion and mobilization are regulated by BM niche cells[3], including osteolineage cells (mature osteoblasts and osteoblast progenitors), macrophages, osteoclasts, endothelial cells, neutrophils, and mesenchymal stem and stromal cells. These BM niche cells can secrete a variety of growth factors or cytokines that affect HSC function[3–7], for examples, osteolineage cells produce granulocyte colony-stimulating factor (G-CSF)[8], the stromal cells that surround HSCs release stem cell factor[9] and endothelial cells produce E-selectin ligand to regulate HSC proliferation[10]. Although HSCs can produce all immune cell lineages in the blood, it is less clear whether signals from the blood affect HSC homeostasis. We propose that extramedullar cytokines in the blood also regulate the BM niche to affect HSC expansion and mobilization.

Leukocyte cell-derived chemotaxin 2 (LECT2) is a multi-functional factor secreted by the liver into the blood[11]. LECT2 is involved in many pathological conditions, such as sepsis[12], diabetes[13], systemic amyloidosis[14,15] and hepatocarcinogenesis[16]. LECT2 activates macrophages via interacting with CD209a (ref. 12), a C-type lectin related to dendritic cell-specific ICAM-3-grabbing non-integrin[17,18], and is mainly expressed in macrophages and dendritic cells[12,19]. In the BM niche, macrophages play an important role in HSC expansion and mobilization[20,21]. Therefore, LECT2 may regulate HSC function via activating BM macrophages.

In this study, we report a previously unknown role of LECT2 in HSC homeostasis and the BM microenvironment. We determine that LECT2 is a novel candidate gene responsible for HSC expansion and mobilization via interacting with CD209a in macrophages and osteolineage cells. The LECT2/CD209a axis affects the expression of tumour necrosis factor (TNF) in macrophages and osteolineage cells, and HSC homeostasis is evaluated in TNF knockout (KO) mice. TNF affects the stromal cell-derived factor-1-CXC–chemokine receptor 4 (SDF-1–CXCR4) axis to regulate HSC homeostasis. We further compare the effects of LECT2 and G-CSF on HSC mobilization. These results describe an extramedullar cytokine that regulates HSC expansion in the BM and mobilization to the blood.

## Results

### LECT2 enhances HSC expansion and mobilization.
We first investigated the relationship between LECT2 expression and HSC number in the blood of humans in steady state. The number of HSCs was positively correlated with plasma LECT2 levels in humans (Fig. 1a). The effect of recombinant LECT2 on mouse HSC homeostasis in vivo was evaluated (Fig. 1b). The number of colony-forming unit cells (CFU-Cs), white blood cells (WBCs) and $Lin^-Sca-1^+c-Kit^+$ (LSK) cells in the blood increased after LECT2 treatment for 5 days (Fig. 1c,d). Moreover, the LECT2 treatment also enhanced the CFU-Cs, WBCs and LSK cells in the blood of C3H/HeJ mice, a strain that is relatively insensitive to endotoxin (Supplementary Fig. 1a–c). In the BM, LECT2 did not affect the number of WBCs, but increased the number of LSK cells after treatment for 3 days (Fig. 1e). Kinetic studies demonstrated that LECT2 increased the number of LSK cells in the blood at 4 and 5 days after treatment, but not at earlier time points (Fig. 1f). This increase of LSK cell number in LECT2-treated mice was accompanied by the increased proliferation of LSK cells (Fig. 1g,h). LECT2 treatment for 3 days also increased the number of BM long-term HSCs (LT-HSCs, LSK $CD34^-$

$Flk2^-$ cells), short-term HSCs (ST-HSCs, LSK $CD34^+Flk2^-$ cells) and lymphoid-primed multipotent progenitors (LMPPs, LSK $CD34^+Flk2^+$ cells; Fig. 1i). Furthermore, the number of CFU-Cs, LSK cells in the blood and LSK cells in the BM decreased in LECT2 KO mice (Fig. 1j–l).

Because the phenotype (LSK $Flt3^-CD34^-$) by flow cytometry is not always reliable when mice are stimulated by an exogenous agent[22–24], we further measured the repopulating activity of LSK cells from the BM of mice treated with LECT2 or PBS for 3 days by monitoring animal survival for 60 days after transplantation with LSK cells. The frequency of reconstituting cells was 1 per 194.1 LSK cells in PBS-treated mice and 1 per 77.3 LSK cells in LECT2-treated mice (Fig. 2a). We further evaluated their reconstitution efficiency using a competitive repopulating assay. The frequency of competitive repopulating units (CRUs) was 1 per 18.4 LSK cells in PBS-treated mice and 1 per 12.0 LSK cells in LECT2-treated mice (Fig. 2b). Normal HSC self-renewal is important for the maintenance of haematopoiesis[25,26]. We further examined the extent of the repopulation of the HSCs and the different lineages in the initial 1:1 ratio of LSK cells from the BM of LECT2-treated C57BL/6 (CD45.2) mice and B6.SJL (CD45.1) mice. We observed higher frequencies of HSCs after the first transplantation with LSK cells from the BM of LECT2-treated mice for 3 days compared with PBS-treated mice (Fig. 2c). We also observed higher repopulation of both lymphoid and myeloid cells in the mice engrafted with LSK cells from the BM of LECT2-treated mice for 3 days compared with PBS-treated mice (Fig. 2d). A repopulating unit (RU) value in the LSK cells from the BM of LECT2-treated mice was 1.43-fold of that of PBS-treated mice (Fig. 2e). To further investigate the long-term self-renewal capacity, we performed second transplantation. LECT2 treatment did not enhance the reconstitution potential after second transplantation (Fig. 2f). Furthermore, the RU value in the BM cells of 3-day LECT2-treated mice was 3.12-fold of that of PBS-treated mice (Fig. 2g).

Because HSCs participate in the inflammatory response during sepsis[27] and LECT2 administration promotes septic mouse survival[12], we further evaluated the effect of LECT2-treated HSCs on sepsis. $Lin^-c-Kit^+$ cells from the BM of LECT2-treated mice promoted survival and decreased the bacterial burden in septic mice (Supplementary Fig. 2a,b). These results reveal that LECT2 regulates HSC expansion, mobilization and transplantation efficiency.

### CD209a mediates the effect of LECT2 on HSC homeostasis.
LECT2 can interact with two receptors, CD209a (ref. 12) and c-MET[28]. We next determined which receptor mediates the effect of LECT2 on HSC homeostasis (Supplementary Fig. 3a). CD209a blockade, but not c-MET inhibition, reduced the effect of LECT2 on the HSC number in the blood and BM (Supplementary Fig. 3b–g). There was no change on HSC number in the blood and BM after c-MET inhibitor treatment alone (Supplementary Fig. 3b,d,e).

The LECT2 receptor was investigated in CD209a KO mice. The effect of LECT2 on HSC homeostasis was abolished in CD209a KO mice (Fig. 3a–d). Moreover, both the number of HSCs and plasma LECT2 levels were reduced in CD209a KO mice (Fig. 3a–e). Anti-LECT2 IgG reduced the number of LSK cells in the blood of the wild type (WT) but not CD209a KO mice (Fig. 3f). However, LECT2 messenger RNA (mRNA) could not be detected in the BM transcriptome, suggesting that LECT2 is an extramedullar cytokine. These data suggest that CD209a mediates the effects of both endogenous and exogenous LECT2 on HSCs.

We also investigated the potential role of LECT2/CD209a signal in regulating HSC regeneration after 5-fluorouracil (5FU)

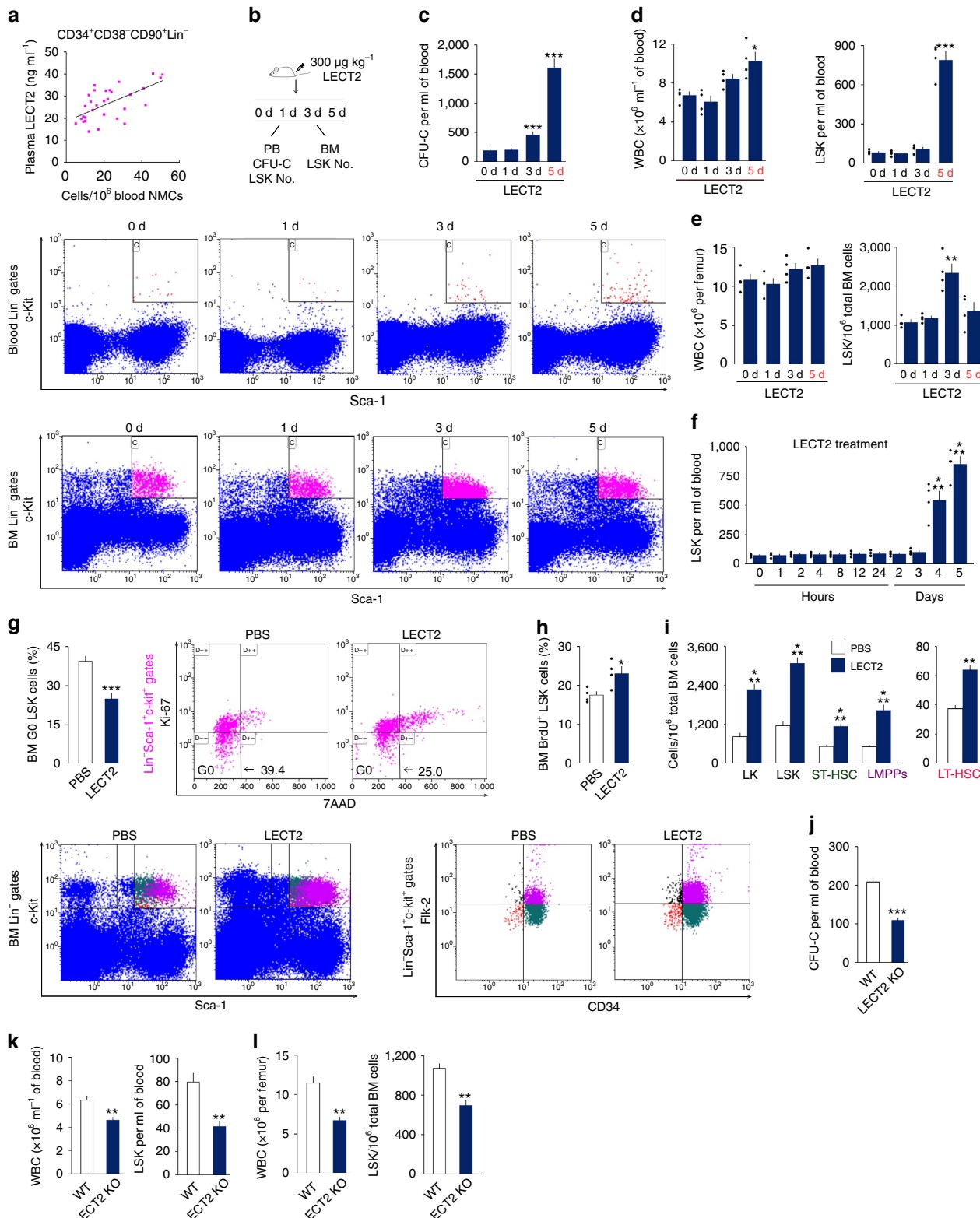

**Figure 1 | LECT2 increases the mobilization and expansion of HSCs and their transplantation potential.** (**a**) The correlation between human plasma LECT2 levels and the number of CD34+CD38−CD90+Lin− cells in the blood (Pearson correlation coefficient = 0.624, $P < 0.001$); $n = 31$. (**b**) The protocol used to evaluate the effect of LECT2 on HSC homeostasis. (**c**) The number of CFU-Cs per millilitre of mouse blood; $n = 10$. (**d**) The number of WBCs and LSK cells in the blood. (**e**) The number of WBCs and LSK cells in BM. (**f**) The number of LSK cells in the blood at different time points after LECT2 administration. (**g,h**) Percentage of quiescent G0 (**g**) and BrdU+ (**h**) LSK cells in the BM of LECT2-treated mice. (**i**) Effects of LECT2 on the number of LK cells, LSK cells, LT-HSCs, ST-HSCs and LMPPs in BM. $n = 5$. (**j–l**) The number of CFU-Cs (**j**), WBCs and LSK cells (**k**) in the blood and the number of WBCs and LSK cells (**l**) in the BM of LECT2 KO mice; $n = 5$. The small black dots in histograms are data points. The data represent means ± s.e.m. The data are representative of two (**d–l**) and three (**a,c**) independent experiments. $*P < 0.05$, $**P < 0.01$, $***P < 0.001$ using Pearson's correlation coefficient analysis (**a**) and one-way analysis of variance (ANOVA) (**c–l**).

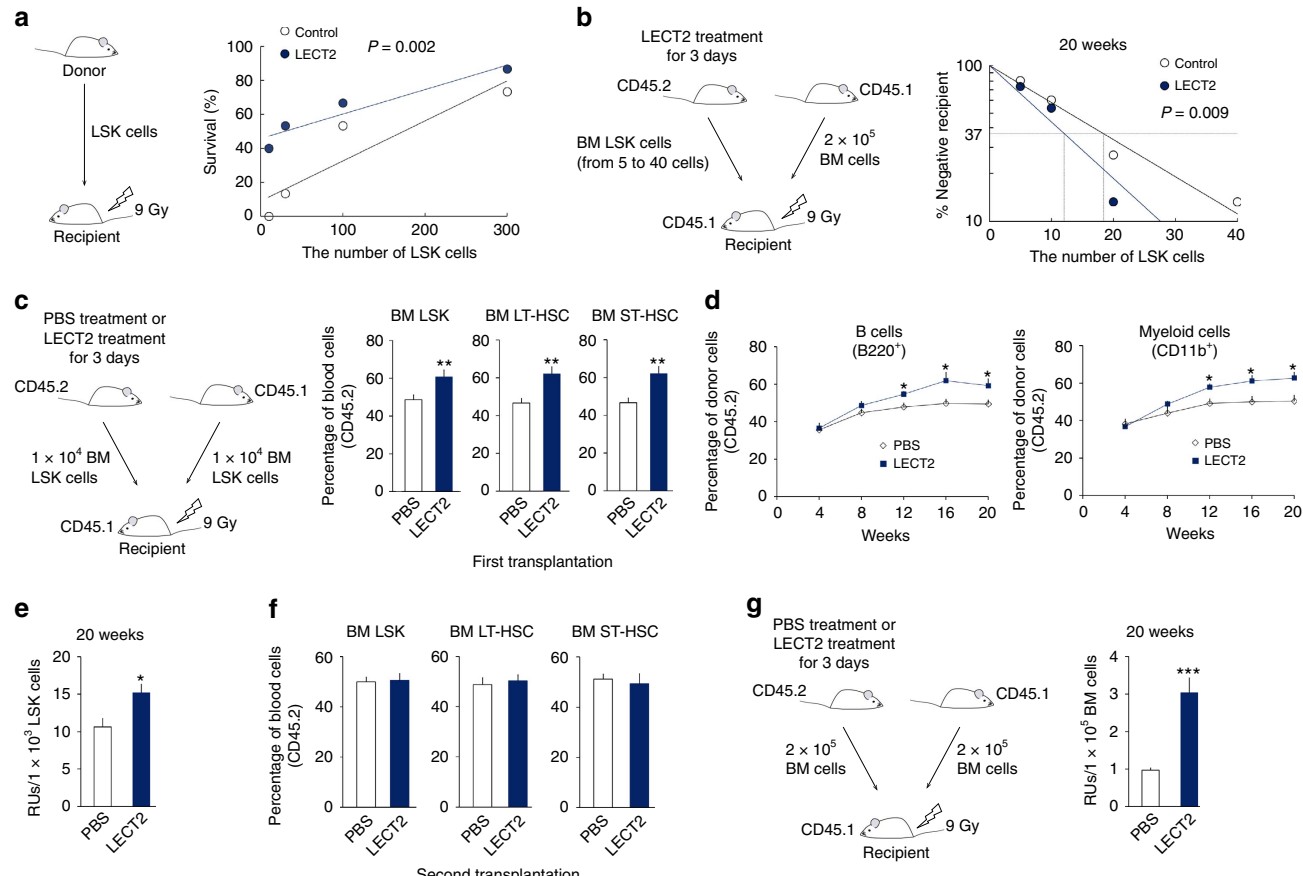

**Figure 2 | LECT2 increases the transplantation potential of BM LSK cells.** (**a**) The survival rate was analysed following transplantation with LSK cells from the BM of 3-day LECT2 treated mice; $n = 15$ each point. (**b**) Competitive repopulation unit (CRU) content within each group of mice competitively transplanted with LSK cells at each dose; $n = 15$ each point. Horizontal dashed line, 37% of recipient mice failed to engraft. Vertical dashed lines, various CRU frequencies for each treatment. (**c**) The percentages of donor LSK cells, LT-HSCs, and ST-HSCs in BM at 20 weeks after first competitive transplantation with PBS- or LECT2-treated mice are shown; $n = 5$. (**d**) The percentages of B cells (B220[+]) and myeloid cells (CD11b[+]) in the peripheral blood of the chimaeric mice competitively transplanted with LSK cells; $n = 6$. (**e**) Repopulating units based on donor chimaerism determined by flow cytometry in peripheral blood 20 weeks after competitive transplantation with LSK cells; $n = 6$. (**f**) The percentages of donor LSK cells, LT-HSCs, and ST-HSCs at 20 weeks after second competitive transplantation with LSK cells. $n = 5$. (**g**) Repopulating units (RU) based on donor chimaerism determined by flow cytometry in peripheral blood 20 weeks after competitive transplantation with BM cells of 3-day LECT2 treated mice; $n = 5$. The data represent means ± s.e.m. The data are representative of two independent experiments. $*P < 0.05$, $**P < 0.01$, $***P < 0.001$ using Poisson's statistic (**a**,**b**) and one-way ANOVA (**c**–**g**).

treatment, which leads to BM stress by reducing the number of HSCs[29]. In CD209a KO mice, there were fewer HSCs in the BM than in WT mice after 5FU treatment (Supplementary Fig. 4a–e). The expression levels of LECT2 and CD209a were downregulated at 4 and 8 days after 5FU treatment, and were subsequently upregulated (Supplementary Fig. 4f,g). Furthermore, LECT2 administration enhanced the regeneration of HSCs in the BM after 5FU treatment (Supplementary Fig. 4h–l). These data suggest that LECT2/CD209a signal participates in HSC regeneration during 5FU stress.

We further aimed to identify the cellular localization of CD209a. CD209a[+] cells exhibited high expression of CD169 and RunX2 (Fig. 3g), markers of macrophages and osteolineage cells, respectively. Furthermore, CD209a was positive in macrophages and osteolineage cells, but negative in LSK cells (Fig. 3h). These data suggest that the LECT2/CD209a signal is targeted to macrophages and osteolineage cells.

**Macrophages and osteolineage cells are the target of LECT2.** We further investigated whether macrophages and osteolineage cells mediate the effect of LECT2 on HSCs. Diphtheria toxin (DT)

treatment in CD169[DTR/+] mice has been used to deplete macrophages[20]. The number of CD169[+] cells decreased in DT-treated CD169[DTR/+] mice compared with WT mice (Fig. 4a). After LECT2 treatment, the number of LSK cells in the BM and blood of DT-treated CD169[DTR/+] mice decreased compared with WT mice (Fig. 4b,c). In DT-treated CD169[DTR/+] mice, the number of LSK cells in the blood slightly increased after LECT2 treatment (Fig. 4c). LSK cells were measured after macrophage depletion by the administration of clodronate liposome. After LECT2 treatment, the number of LSK cells in the BM and blood of clodronate liposome-treated mice decreased compared with PBS liposome-treated mice (Fig. 4d–f).

To assess the role of osteolineage cells in the effect of LECT2 on HSCs, biglycan (Bgn) KO mice, in which osteolineage cells are reduced, were used[30]. The number of LSK cells in the BM and blood decreased in Bgn[−/0] mice compared with WT mice after LECT2 treatment (Fig. 5a–c). Strontium chloride (SrCl$_2$) has been used to increase osteolineage cells[31], and we used SrCl$_2$ to confirm the role of osteolineage cells in the LECT2 effect. The number of LSK cells in the SrCl$_2$-treated mice increased compared with the PBS-treated mice after LECT2 treatment (Supplementary Fig. 5).

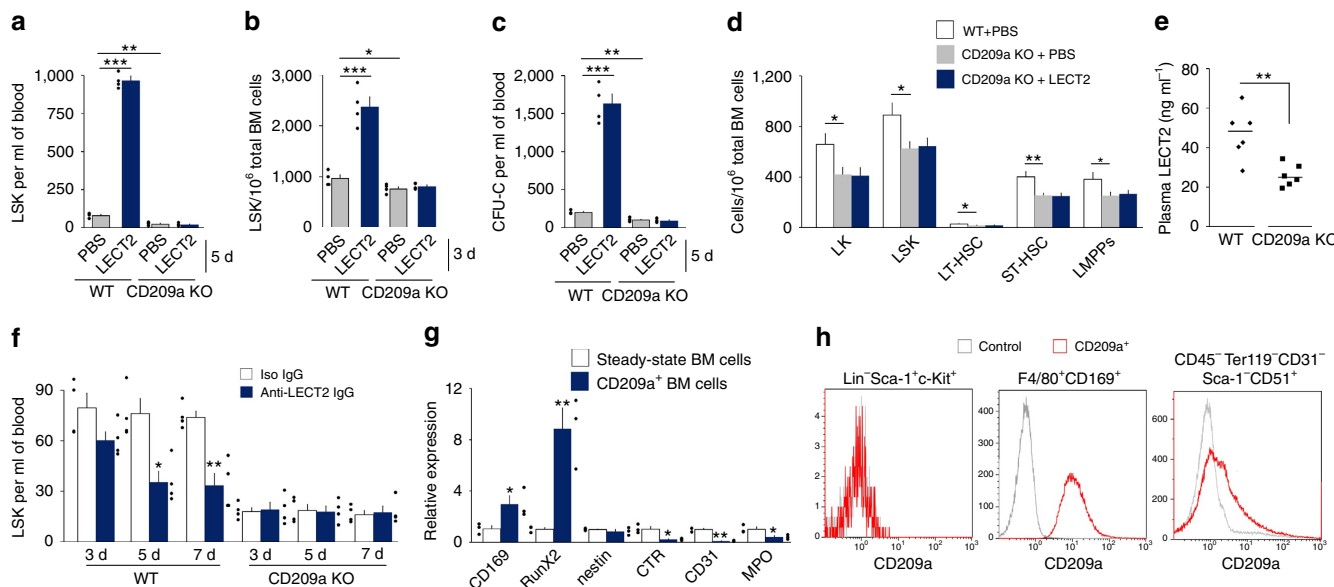

**Figure 3 | CD209a in BM niche cells mediates the effect of LECT2 on HSC mobilization and expansion.** (**a,b**) The number of LSK cells in the blood (**a**) and BM (**b**) of WT and CD209a KO mice after PBS or LECT2 treatment. (**c**) The number of CFU-Cs in the blood of WT and CD209a KO mice. (**d**) The number of LK cells, LSK cells, LT-HSCs, ST-HSCs and LMPPs in the BM of LECT2-treated CD209a KO mice; $n = 4$. (**e**) LECT2 protein levels in WT and CD209a KO mice; $n = 6$. (**f**) LSK cell number in WT and CD209a KO mice after isoIgG or anti-LECT2 IgG treatment. (**g**) The mRNA levels of BM cell markers in CD209a[+] BM cells relative to the level in steady-state BM cells. CD169, a marker of macrophages; RunX2, a marker of osteolineage cells; nestin, a marker of mesenchymal stem or stromal cells; calcitonin receptor (CTR), a marker of osteoclasts; CD31, a marker of endothelial cells; myeloperoxidase (MPO), a marker of neutrophils. (**h**) A representative histogram of the percentage of CD209a-positive cells within LSK cells, macrophages and osteolineage cells in the steady-state condition; $n = 4$. The small black dots in histograms are data points. The data represent means ± s.e.m. The data are representative of two independent experiments. $^*P < 0.05$, $^{**}P < 0.01$, $^{***}P < 0.001$ using one-way ANOVA.

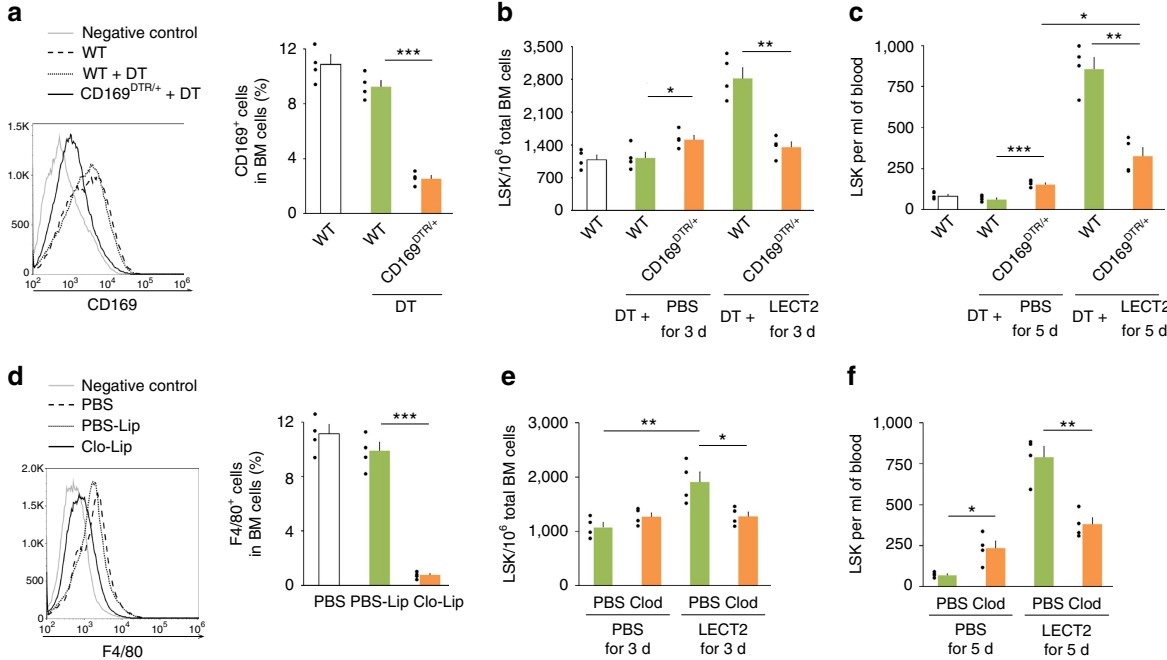

**Figure 4 | Macrophages mediate the effect of LECT2 on HSC expansion and mobilization.** (**a**) The number of CD169[+] cells in WT mice and WT or CD169[DTR/+] mice injected with DT. (**b**) The number of LSK cells in the BM of WT or CD169[DTR/+] mice after PBS or LECT2 treatment. (**c**) The number of LSK cells in the blood of WT or CD169[DTR/+] mice after PBS or LECT2 treatment. (**d**) Effect of PBS liposomes (PBS-Lip) and clodronate liposomes (Clo-Lip) on macrophage number. (**e**) Effect of LECT2 on the number of BM LSK cells after macrophage depletion. (**f**) Effect of LECT2 on the number of blood LSK cells after macrophage depletion; $n = 4$. The small black dots in histograms are data points. The data represent means ± s.e.m. The data are representative of two independent experiments. $^*P < 0.05$, $^{**}P < 0.01$, $^{***}P < 0.001$ using one-way ANOVA.

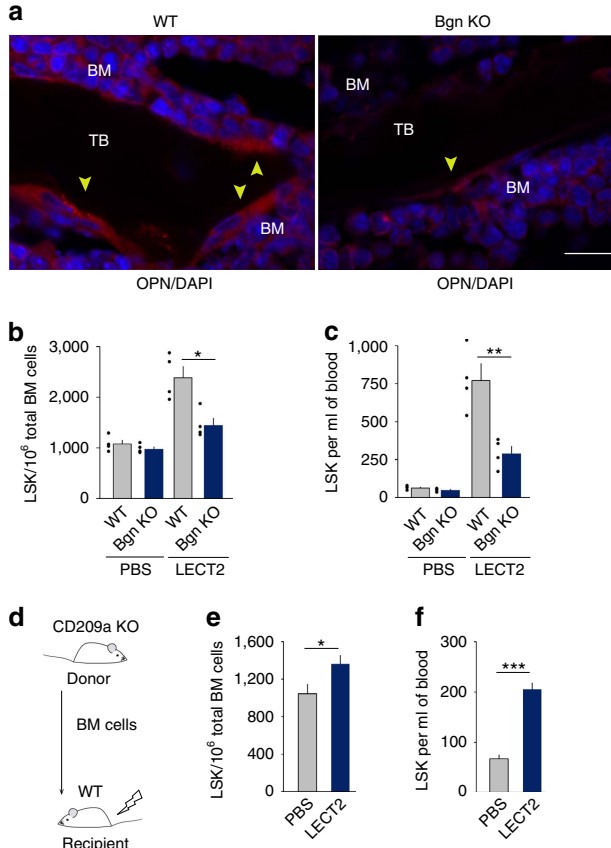

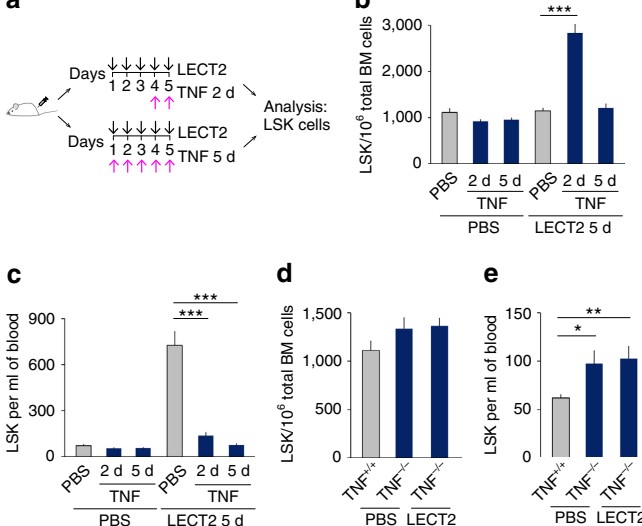

**Figure 6 | TNF mediates the effect of LECT2 on HSC expansion and mobilization.** (**a**) The protocol for TNF and LECT2 treatment. (**b**) The number of LSK cells in the BM of mice treated with LECT2 combined with TNF. (**c**) The number of LSK cells in the blood of mice treated with LECT2 combined with TNF. $n = 5$. (**d,e**) LSK cells in the BM and blood of TNF KO mice after LECT2 treatment; $n = 6$. The data represent means ± s.e.m. The data are representative of two (**b,c**) and three (**d,e**) independent experiments. *$P < 0.05$, **$P < 0.01$, ***$P < 0.001$ using one-way ANOVA.

**Figure 5 | Osteolineage cells partly mediate the effect of LECT2 on HSC expansion and mobilization.** (**a**) Representative tissue expression of OPN in the BM of WT and Bgn$^{-/0}$ mice. Anatomical landmarks are indicated: bone marrow (BM) and trabecular bone (TB). The OPN positive cells are indicated by arrow heads. The scale bar represents 20 μm. (**b**) The number of LSK cells in the BM of WT or Bgn$^{-/0}$ mice after PBS or LECT2 treatment. (**c**) The number of LSK cells in the blood of WT or Bgn$^{-/0}$ mice after PBS or LECT2 treatment. (**d**) The protocol used to produce chimaeras of WT mice with BM cells from CD209a KO mice. (**e**) Effect of LECT2 on LSK cell number in the BM of WT mice with BM cells from CD209a KO mice. (**f**) Effect of LECT2 on LSK cell number in the blood of WT mice with BM cells from CD209a KO mice; $n = 4$. The small black dots in histograms are data points. The data represent means ± s.e.m. The data are representative of two independent experiments. *$P < 0.05$, **$P < 0.01$, ***$P < 0.001$ using one-way ANOVA.

Furthermore, BM cells from CD209a KO mice were injected intravenously (i.v.) into lethally irradiated (9 Gy) WT mice that expressed CD209a in osteolineage cells but not in macrophages (Fig. 5d). In these mice, LECT2 still led to the increase of LSK cell number in the BM and blood, but only a 1.3-fold and 3.1-fold increase in the BM and blood, respectively (Fig. 5e,f).

**TNF secretion is suppressed by the LECT2 signal.** Because LSK cells did not express CD209a (Fig. 3h), we hypothesized that secreted soluble factors from macrophages or osteolineage cells might mediate the effect of LECT2. The data from transcriptome sequencing revealed that several cytokines were downregulated in the BM after LECT2 treatment (Supplementary Fig. 6a). Moreover, the protein levels of TNF and CC motif chemokine ligand (CCL)3 were downregulated to <25% of the control levels in the BM supernatant of LECT2-treated mice (Supplementary Fig. 6b).

TNF plays an important role in regulating HSC expansion and emergence[27,32,33]. CCL3 has also been identified as a stem cell inhibitory factor[34]. TNF mediated the downregulation of CCL3 expression by LECT2 in the macrophages (Supplementary Fig. 7a), suggesting that TNF mediates the effect of LECT2/CD209a signal on CCL3 downregulation. Therefore, we hypothesized that TNF played important roles in LECT2/CD209a signal on HSCs in the BM. The expression levels of TNF were not upregulated in the BM of CD209a KO mice after LECT2 treatment (Supplementary Fig. 7b). Because CD209a was mainly expressed in macrophages and osteolineage cells (Fig. 3h), we further investigated the effect of LECT2/CD209a signal on TNF expression levels *in vitro*. TNF levels in both osteolineage cells and macrophages were downregulated after LECT2 treatment (Supplementary Fig. 7c).

**TNF mediates the effect of LECT2 on HSCs.** We further investigated whether TNF participated in HSC homeostasis. Recombinant TNF was employed to investigate the role of TNF in the effect of LECT2 (Fig. 6a). TNF inhibited the induction of HSC mobilization and expansion by LECT2 (Fig. 6b,c). Furthermore, the effect of LECT2 on HSC expansion and mobilization was abolished in TNF KO mice (Fig. 6d,e). We further analysed HSC mobilization in nonobese diabetic/severe combined immuno-deficient (NOD–SCID) mice using human HSCs after the administration of LECT2 and TNF, either alone or in combination. TNF treatment blocked the LECT2-induced mobilization of mature human CD45$^+$ cells and CD34$^+$ CD38$^-$ CD90$^+$ Lin$^-$ cells in NOD–SCID mice that received human cord blood mononuclear cells (Supplementary Fig. 8a,b). These data demonstrate that TNF mediates the LECT2-induced mobilization of both human and murine HSCs.

**TNF disrupts the SDF-1–CXCR4 axis.** The SDF-1–CXCR4 axis is the major regulator of HSC migration[8]. We observed that CXCR4 blockade increased HSC mobilization following LECT2 induction (Fig. 7a,b). LECT2 downregulated the surface expression of CXCR4 in LSK cells, and this downregulation was

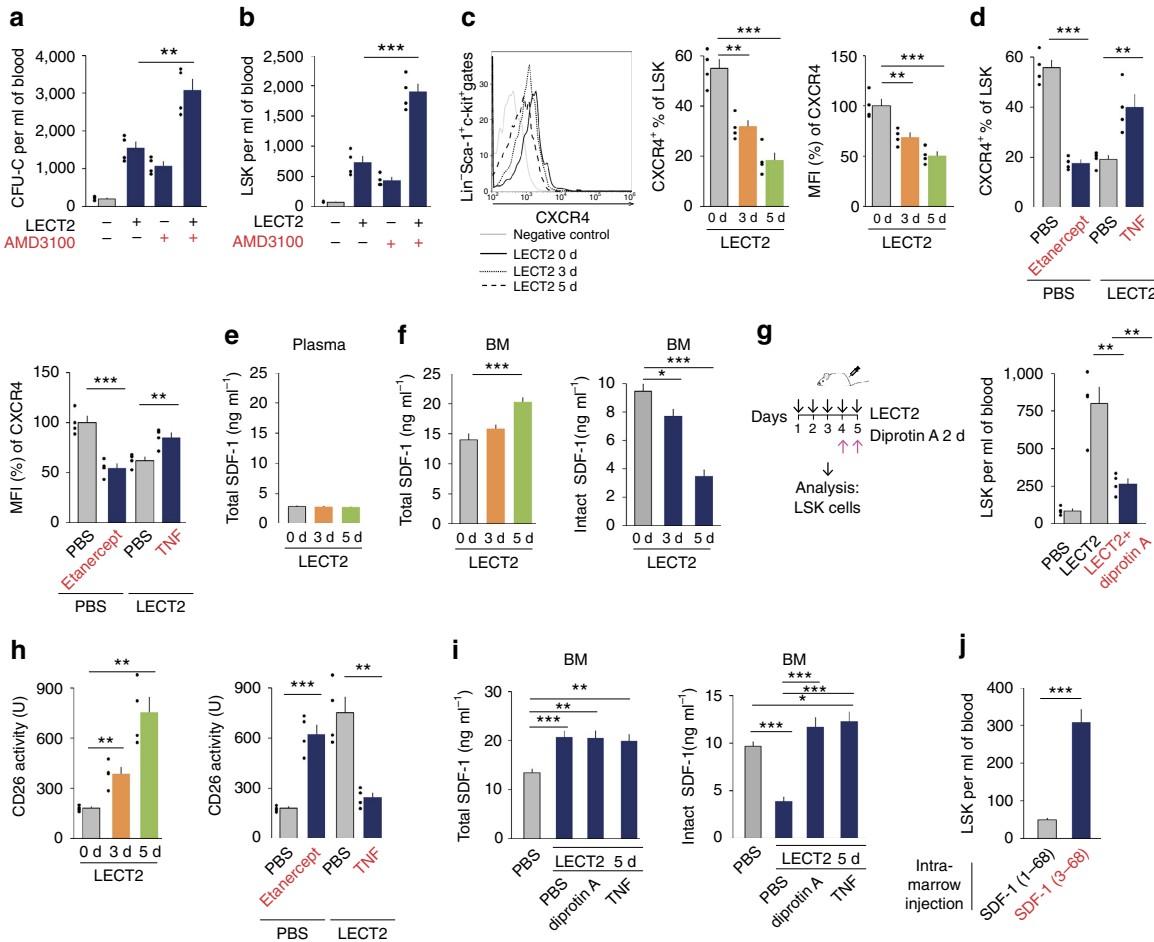

**Figure 7 | TNF mediates the effect of LECT2 on the SDF-1-CXCR4 axis.** (**a**) Effect of CXCR4 inhibitor (AMD3100) on the number of blood CFU-Cs after LECT2 treatment. (**b**) Quantification of the number of blood LSK cells in mice injected with AMD3100 after LECT2 treatment; $n = 4$. (**c**) Surface CXCR4 expression on LSK cells from flushed BM cells after LECT2 treatment. $n = 4$. (**d**) Surface CXCR4 expression on LSK cells from the BM of mice after TNF or etanercept treatment combined with LECT2 induction for 3 days; $n = 4$. (**e**) SDF-1 protein levels in the plasma of PBS- or LECT2-treated mice; $n = 10$. (**f**) Total and intact SDF-1 in BM after LECT2 treatment; $n = 5$. (**g**) The number of LSK cells in the blood were analysed after LECT2 and/or diprotin A treatment. (**h**) CD26 peptidase activity in flushed BM cells after TNF or etanercept treatment combined with LECT2 induction for 3 days; $n = 4$. (**i**) Total and intact SDF-1 in BM after diprotin A or TNF treatment combined with LECT2 incubation; $n = 5$. (**j**) The number of LSK cells in the blood was analysed after intra-femur SDF-1 (1–68) or SDF (3–68) injection; $n = 5$. The small black dots in histograms are data points. The data represent means ± s.e.m. The data are representative of two (**a**–**d** and **g**–**j**) and three (**e**,**f**) independent experiments. *$P < 0.05$, **$P < 0.01$, ***$P < 0.001$ using one-way ANOVA.

mediated by TNF (Fig. 7c,d). LECT2 slightly upregulated total SDF-1 in the BM and did not alter the expression of total SDF-1 in the plasma (Fig. 7e,f). Furthermore, LECT2 treatment led to the downregulation of intact SDF-1 (1–68; Fig. 7f). Because CD26 can remove the first two amino-acid residues of intact SDF-1 to produce SDF-1 (3–68), an antagonist of chemotaxis[35], we further investigated the role of CD26 in LECT2 signal. The effect of LECT2 on the mobilization of LSK cells was suppressed by a CD26 inhibitor diprotin A[36], which was subcutaneous (s.c.) injected 5 μmol twice per day (Fig. 7g). CD26 activity in the BM was upregulated by LECT2 but inhibited by TNF (Fig. 7h). LECT2 upregulated total SDF-1 and downregulated intact SDF-1 in the BM (Fig. 7i). In LECT2-treated mice, total SDF-1 levels did not change after diprotin A or TNF treatment, whereas intact SDF-1 was upregulated after diprotin A or TNF treatment (Fig. 7i). These results suggest that LECT2 increases CD26 activity to reduce intact SDF-1 via downregulation of TNF. Because the total SDF-1 level was upregulated in the BM, the downregulation of intact SDF-1 suggested that SDF-1 (3–68) was upregulated in the BM after LECT2 treatment. Next, we used SDF-1 (1–68) and SDF-1 (3–68) to investigate HSC mobilization.

The number of LSK cells increased in the blood after intra-marrow injection of SDF-1 (3–68) compared with SDF-1 (1–68) (Fig. 7j).

**The effects of G-CSF and LECT2 on HSC homeostasis.** G-CSF is the most widely used HSC-mobilizing agent in clinical settings[8]. G-CSF and LECT2 did not have a synergistic effect on HSC expansion and mobilization (Fig. 8a–e). To further investigate the differences in HSC mobilization due to LECT2 and G-CSF in detail, LT-HSCs and LMPPs in the blood were both measured. The number of LT-HSCs and LMPPs in the blood both increased after LECT2 or G-CSF treatment (Fig. 8f). Furthermore, the effect of G-CSF on LSK cell mobilization was completely lost 20 days after irradiation (6 Gy; Fig. 8g). Twenty days after irradiation, the effect of LECT2 on HSC expansion and mobilization in mice was evaluated. LECT2 increased the number of LSK cells and WBCs in the BM of irradiated mice (Fig. 8h). LECT2 also markedly increased the number of CFU-Cs, LSK cells and WBCs in the blood of irradiated mice (Fig. 8i,j). By contrast, G-CSF did not induce a change in the number of LSK cells in the

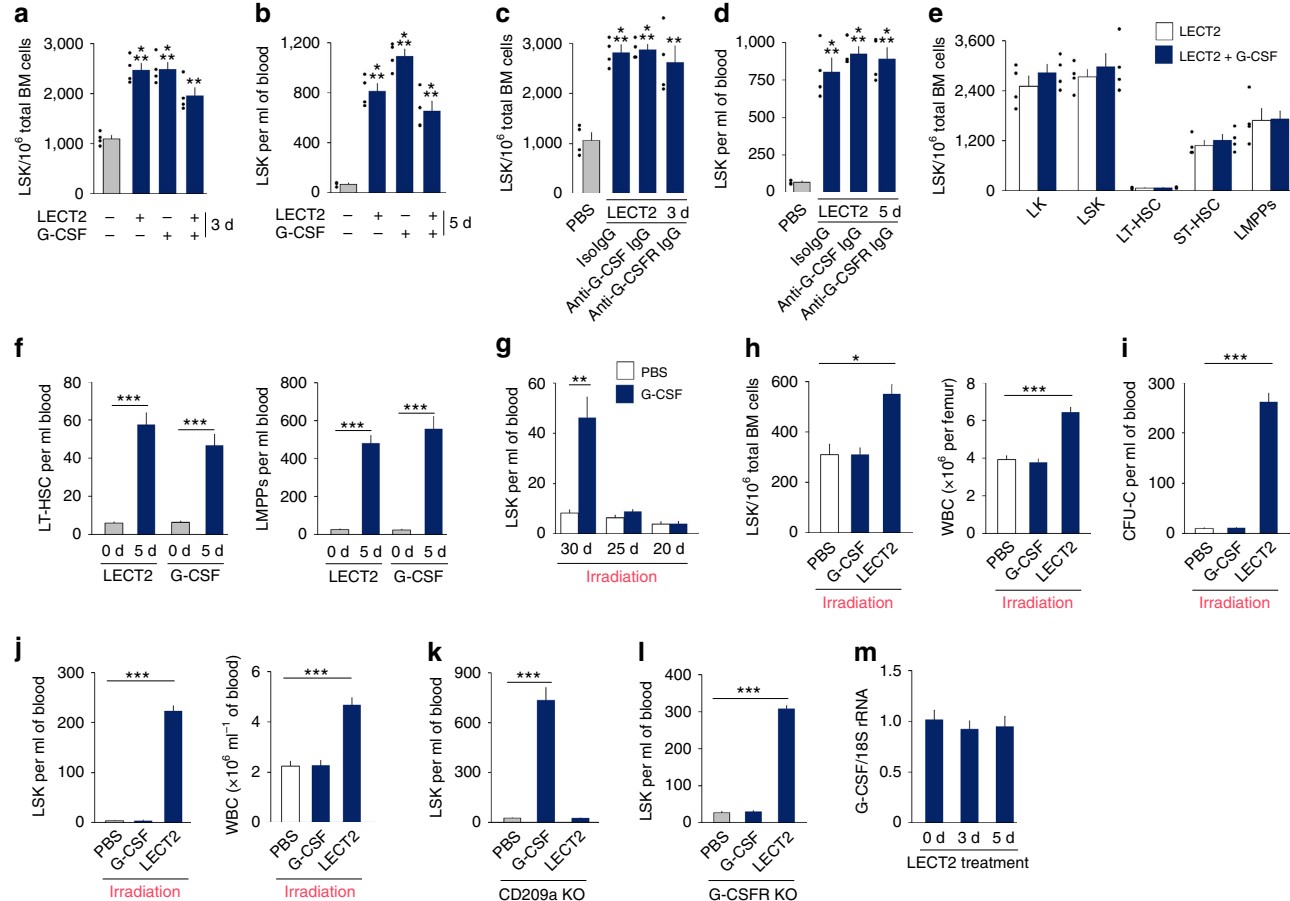

**Figure 8 | LECT2 and G-CSF have independent effects on HSCs.** (**a,b**) The number of LSK cells in BM or blood after PBS, LECT2, G-CSF or LECT2 + G-CSF treatment. (**c,d**) LSK cell number in the BM or blood of mice treated with PBS or LECT2 combined with isoIgG, anti-G-CSF IgG or anti-G-CSFR IgG. (**e**) The number of LK cells, LSK cells, LT-HSCs, ST-HSCs and LMPPs in BM following LECT2 or LECT2 + G-CSF treatment; $n = 4$. (**f**) The number of LT-HSCs and LMPPs in the blood following LECT2 or G-CSF treatment. (**g**) LSK cell number in irradiated mice (6 Gy) after G-CSF treatment at different time points. (**h**) The number of LSK cells and WBCs in the BM of irradiated mice 20 days after G-CSF or LECT2 treatment. (**i,j**) The number of CFU-Cs, LSK cells and WBCs in the blood of irradiated mice 20 days after G-CSF or LECT2 treatment. (**k**) The effect of G-CSF or LECT2 on the number of LSK cells in CD209a KO mice. (**l**) The effect of G-CSF or LECT2 on the number of LSK cells in G-CSFR KO mice; $n = 5$. (**m**) mRNA expression of G-CSF after LECT2 treatment. The small black dots in histograms are data points. The data represent means ± s.e.m. The data are representative of two independent experiments. $*P < 0.05$, $**P < 0.01$, $***P < 0.001$ using one-way ANOVA.

irradiated mice (Fig. 8h–j). To further determine whether the effects of LECT2 and G-CSF were independent, we administered G-CSF to CD209a KO mice and LECT2 to G-CSF receptor (G-CSFR) KO mice. In CD209a KO mice, G-CSF treatment markedly increased the number of LSK cells in the blood, whereas LECT2 treatment had no effect on LSK cell mobilization (Fig. 8k). In G-CSFR KO mice, LECT2 treatment markedly increased the number of LSK cells in the blood, whereas G-CSF treatment had no effect on LSK cell mobilization (Fig. 8l). Moreover, mRNA expression of G-CSF in the BM was not altered by LECT2 treatment (Fig. 8m). LECT2 mRNA expression was not detected in the BM after G-CSF treatment.

We further analysed the number of reconstituting HSCs mobilized per ml blood by LECT2 and G-CSF. A RU value per ml blood from LECT2-treated mice was 1.86-fold of that of G-CSF-treated mice (Fig. 9a). Using limiting dilution competitive transplantation assays, we could demonstrate that LECT2 mobilized more CRUs than G-CSF (Fig. 9b). We further investigated the effect of G-CSF on the LECT2 downstream signal, target cells and TNF. LECT2 treatment increased the number of macrophages but did not change the number of osteolineage cells, whereas G-CSF downregulated both

macrophages (F4/80$^+$CD169$^+$) and osteolineage cells (CD45$^-$Ter119$^-$CD31$^-$Sca-1$^-$CD51$^+$) in the BM (Fig. 9c,d). Both LECT2 and G-CSF downregulated TNF production in the BM (Fig. 9e). However, BM TNF level in LECT2-treated mice was lower than that in G-CSF-treated mice after 5 days of administration (Fig. 9e). Moreover, the marked effect of G-CSF on the expansion and mobilization of LSK cells was maintained in TNF KO mice (Fig. 9f,g). These results reveal that the effects of LECT2 and G-CSF on the BM niche are different.

**Discussion**

In the present study, we demonstrated that recombinant LECT2 injection induced HSC expansion and mobilization. In addition, we determined that CD209a, the LECT2 receptor, is mainly expressed in macrophages and osteolineage cells, which mediate the effect of LECT2 on HSC homeostasis. The number of CFU-Cs and LSK cells both decreased in LECT2 KO mice, supporting the connection between HSC homeostasis and LECT2. LECT2 mRNA was not detected in BM in our transcriptome analysis. A previous study showed that LECT2 is secreted by the liver into the blood[11], therefore, LECT2 is an extramedullar cytokine for HSC regulation. Plasma LECT2 levels are downregulated in septic

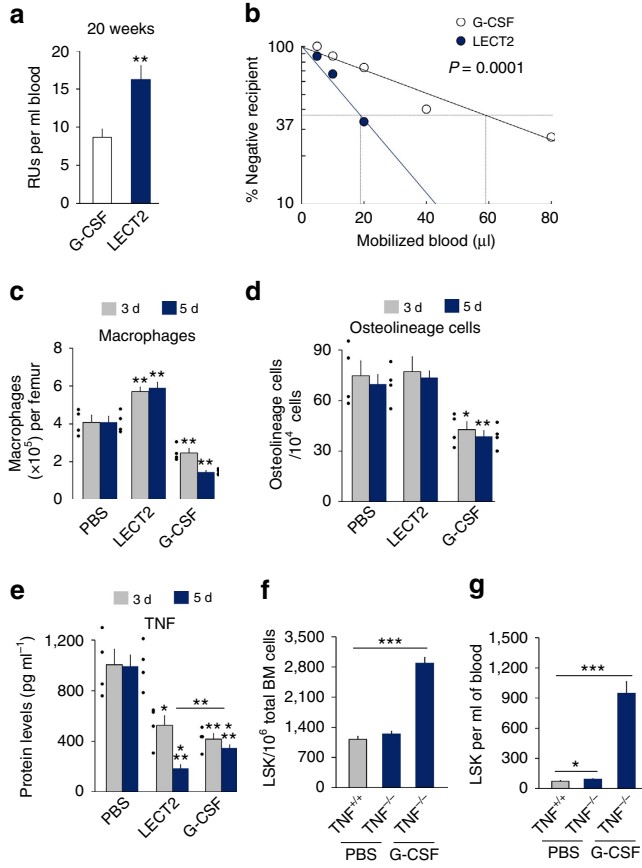

**Figure 9 | G-CSF and LECT2 exhibit different effects on mobilization of RU and CRU, BM cells and TNF expression.** (**a**) RU based on donor chimaerism determined by flow cytometry in peripheral blood 20 weeks after competitive transplantation with blood from 5-day LECT2- or G-CSF-treated mice; $n = 5$. (**b**) CRU frequency in LECT2 or G-CSF mobilized blood was compared by limiting dilution transplantation assays. Engraftment levels of CD45.2 cells were quantified 20 weeks after transplantation; mice with >1% CD45.1 cells were considered positive for mobilized blood engraftment. Dose-dependent engraftment derived from mobilized blood was observed. The CRU frequency was 52.9 ml$^{-1}$ for LECT2 mobilized blood, 16.9 ml$^{-1}$ for G-CSF mobilized blood; $n = 15$ each point. (**c**) The number of macrophages after G-CSF or LECT2 injection. The macrophages were F4/80$^+$CD169$^+$ cells; $n = 4$. (**d**) The number of osteolineage cells after G-CSF or LECT2 injection. PBS and G-CSF, $n = 4$; LECT2, $n = 6$. The osteolineage cells were CD45$^-$Ter119$^-$CD31$^-$Sca-1$^-$CD51$^+$ cells isolated from BM. (**e**) TNF expression after G-CSF or LECT2 injection. 3 d, $n = 4$; 5 d: PBS, $n = 4$; LECT2, $n = 8$; G-CSF, $n = 8$. (**f,g**) The effect of G-CSF on the expansion and mobilization of LSK cells in TNF KO mice; $n = 5$. The small black dots in histograms are data points. The data represent means ± s.e.m. The data are representative of two (**a,b,f,g**) and three (**c–e**) independent experiments. *$P < 0.05$, **$P < 0.01$, ***$P < 0.001$ using one-way ANOVA (**a,c–g**) and Poisson's statistic (**b**).

patients[12]. We previously observed that LECT2 improves sepsis survival in mice by enhancing protective immunity but not by suppressing early inflammation[12]. In the present study, our data confirmed that LECT2-mobilized HSCs improve the outcome of sepsis. These results suggest that HSC mobilization into blood by LECT2 partly explains its function in enhancing protective immunity in sepsis. LECT2 is also involved in a variety of other immune disorders, such as diabetes[13], systemic amyloidosis[14,15] and hepatocarcinogenesis[16]. Therefore, the regulation of HSC homeostasis by LECT2 from the blood may be a common mechanism for maintaining a healthy immune system.

Flow cytometric analysis showed that LECT2 treatment increased the number of LSK cells, LT-HSCs and ST-HSCs in the BM of mice. Because some exogenous agents could change the expression of HSC markers[22–24], we further detected stem cell activity and number (RUs and CRUs) of LSK cells from the BM of LECT2-treated mice. Both the stem cell activity and number of LSK cells from the BM of LECT2-treated mice increased compared with PBS-treated mice, suggesting that LECT2 induces the expansion of HSCs in the BM. Limiting dilution competitive transplantation assays with LECT2-mobilized blood showed the presence of CRUs. Therefore, the LECT2 induces the mobilization of true HSCs, which is important when clinical transplantation of LECT2-mobilized HSCs is planned. Moreover, The number of CRUs in per ml blood of LECT2-treated mice was higher than G-CSF-treated mice. Since the number of LSK cells in the blood of G-CSF- and LECT2-treated mice was similar, the cells with the LSK phenotype mobilized by G-CSF may have reduced haematopoietic potential, which is consistent with previous findings demonstrating G-CSF-mobilized LSK cells have impaired engraftment potential[37].

HSC expansion and mobilization after LECT2 treatment were downregulated in both CD169$^{DTR/+}$ mice treated with DT and Bgn$^{-/0}$ mice compared with WT mice. The effect of LECT2 on HSC homeostasis was reduced but not completely impaired in irradiated mice transplanted with CD209a-deficient HSCs, in which osteolineage cells but not macrophages expressed CD209a. This result suggests that osteolineage cells participate in the effect of LECT2 on HSC homeostasis. Osteolineage cells play an important role in BM niches to regulate HSC homeostasis[3,38]. Some haematopoietic growth factors are produced by osteolineage cell, such as G-CSF[39], a hepatocyte growth factor[40]. In the present study, we observed that TNF was also produced by osteolineage cells to suppress HSC expansion and mobilization. Our data contribute to the elucidation of the complex signals from osteolineage cells that regulate HSC homeostasis.

Macrophages play an important role in HSC expansion and mobilization in the BM niche. Macrophages are pivotal to maintaining the BM niche and regulating the egress of HSCs into the blood[41,42]. The depletion of CD169$^+$ macrophages induces HSC egress by regulating SDF-1 expression[20]. α-Smooth muscle actin$^+$ macrophages secrete prostaglandin E$_2$ to prevent HSC exhaustion[21]. The liver X receptor in macrophages is essential for the rhythmic egress of HSCs into the circulation[43]. We observed that the effect of LECT2 was reduced in CD169$^{DTR/+}$ mice treated with DT compared with WT mice, confirming the important role of macrophages in HSC homeostasis. Moreover, although macrophage depletion induced HSC mobilization, LECT2 also induced HSC mobilization but increased the macrophage number in BM. Therefore, the specific factor secreted by macrophages but not the macrophage number is the essential factor in regulating HSC homeostasis. Transcriptome analysis revealed that TNF, which is constitutively expressed in BM, was markedly downregulated after LECT2 treatment. We subsequently confirmed that TNF mediated the effect of LECT2 on HSC homeostasis. In addition to TNF, oncostatin M was recently suggested to be a macrophage-derived factor mediating HSC function[44]. Thus, macrophages in BM appear to express a variety of signals for regulating of HSC homeostasis.

Macrophages are first identified as the source of TNF after endotoxin activation[45]. TNF is a repressor of HSC activity in BM and may be a mediator of BM failure syndromes after inflammatory diseases[33]. We observed that macrophages and osteolineage cells in BM produced TNF at a concentration of 1 ng ml$^{-1}$ in the supernatant and that TNF mediated the effect of

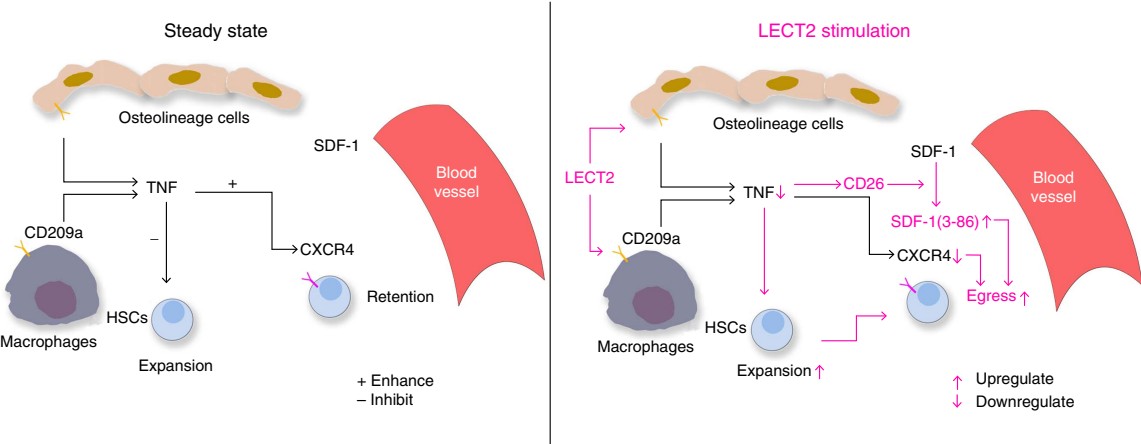

**Figure 10 | LECT2 regulates HSC expansion and mobilization.** In the steady state, macrophages and osteolineage cells secrete TNF in BM. TNF upregulates CXCR4 expression in HSCs and inhibits the expansion of HSCs. After LECT2 stimulation, TNF secretion from CD209a+ macrophages and osteolineage cells is downregulated in BM. TNF downregulation results in the activation of CD26, which cleaves SDF-1 to SDF-1 (3–86). The SDF-1 (3–86) in BM increases HSC egress to the blood. TNF downregulation also reduces CXCR4 expression in HSCs to enhance HSC egress. Moreover, TNF downregulation causes HSC expansion.

LECT2 on HSC homeostasis. Therefore, TNF is a HSC regulator not only under inflammatory conditions but also under steady-state conditions. Although we observed that TNF downregulation contributed to HSC homeostasis, the HSCs in the BM of TNF KO mice were similar to those in WT mice. Furthermore, the percentages of HSCs in the BM of the two types of TNF receptor KO mice are similar to those in WT mice[33]. Normal HSC number in mice that are deficient in TNF or its receptors may result from TNF superfamily of TNF proteins consisting of 19 members and 29 receptors[46]. In the present study, in TNF KO mice, the effect of LECT2 on HSC homeostasis was markedly reduced, suggesting that TNF is the specific downstream target of LECT2 signal. These data illustrate a new pathway for LECT2 signal regulating HSC homeostasis via regulation of the pro-inflammatory cytokine TNF in the absence of infection.

The SDF-1-CXCR4 axis plays a unique role in HSC migration[8]. In the present study, LECT2 decreased CXCR4 expression in LSK cells and increased CD26 activity to cleave intact SDF-1 to SDF-1 (3–68). G-CSF increases plasma SDF-1 and surface expression of CXCR4 in HSCs[8]. The mechanisms of HSC mobilization appear to be different between LECT2 and G-CSF. In G-CSF-induced mobilization, SDF-1 is higher in the plasma than in the BM supernatant. High CXCR4-expressing HSCs are mobilized to the blood. In LECT2-induced mobilization, SDF-1 is also higher in the BM supernatant than in the plasma. The downregulation of CXCR4 in LSK cells after LECT2 treatment mobilized HSCs to blood. Moreover, SDF-1 (3–68) acts as an antagonist of the SDF-1–CXCR4 axis, resulting in a reduction of the migratory response to normal SDF-1 (ref. 36). We observed that SDF-1 (3–68) injection in BM enhanced HSC mobilization. These results demonstrate that LECT2 has different effects on the SDF-1–CXCR4 axis compared with G-CSF.

As a BM niche-derived cytokine, G-CSF may fail to mobilize HSCs in some patients due to several factors, such as prior myelosuppressive chemotherapy, prior radiotherapy[47,48] and diabetes[49]. This failure may be due to severe G-CSF-induced alterations of the BM microenvironment[41,50]. As an extra-medullar cytokine, the HSC-mobilizing effect of LECT2 is similar to that of G-CSF in healthy mice. However, in contract to the loss of the HSC-mobilizing effect of G-CSF in irradiated mice, LECT2 still induces HSC expansion and mobilization in irradiated mice. LECT2 increased the number of macrophages in our study, whereas G-CSF reduced the number of macrophages

in BM[41,50]. Macrophage proliferation by LECT2 induction may enhance HSC expansion and mobilization under myelosuppressive conditions. Furthermore, LECT2 in combination with AMD3100, a CXCR4 antagonist of HSC mobilization[5], exhibited a stronger effect on HSC mobilization compared with LECT2 alone. Therefore, LECT2 may represent a potential HSC-mobilizing agent in patients who do not exhibit mobilization with G-CSF treatment.

In summary, our study reveals the potential value of the clinical application of LECT2 for HSC mobilization. LECT2 can regulate HSC expansion and mobilization by modulating TNF expression in CD209a-expressing osteolineage cells and macrophages (Fig. 10). Because LECT2 is a cytokine that originates in the blood and has been implicated in a variety of immune disorders, our study suggests that immune signals from the blood control HSC expansion and mobilization.

## Methods
**Mice.** Six- to eight-week-old male C57BL/6 mice were purchased from the Zhejiang Province Experimental Animal Center. All injections and measurements were performed in males unless otherwise indicated. B6.SJL, NOD–SCID and C3H/HeJ mice were purchased from Charles River Laboratories (Beijing, China). B6 background CD209a KO mice (B6(FVB)-Cd209atm1.1Cfg/Mmucd) were purchased from the Mutant Mouse Resource Regional Center (University of Missouri, Columbia, MO). CD209a−/− mice were used in the experiments, and their wild-type CD209a+/+ littermates were used as controls. CD169-diphtheria toxin receptor (DTR) heterozygous (CD169DTR/+) mice on a C57BL/6 background[51], which were generated with a DTR complementary DNA[52], were bred in-house by crossing the CD169DTR/DTR mice with C57BL/6 mice. Bgn−/0 mice (B6;129S4(C3)-Bgntm1Mfy/Mmmh) were obtained from Mutant Mouse Resource Regional Center. TNF KO mice (B6;129S-Tnftm1Gkl/J) and G-CSFR KO mice (B6.129X1(Cg)-Csf3rtm1Link/J) were obtained from Jackson Laboratories (Bar Harbor, ME). A mouse sepsis model was established by intraperitoneally (i.p.) injecting of $5 \times 10^6$ CFU of live *Pseudomonas aeruginosa* (ATCC 27853). Survival was monitored once per 12-h period for 96 h. For bacterial clearance analysis, liver, spleen and blood samples were taken from mice after killing. Homogenates of tissues and blood were then serially diluted to determine colony-forming units. The experimental conditions and procedures were approved by the Ningbo University Institutional Animal Care and Use Committee and were consistent with the National Institutes of Health Guide for the Care and Use of Laboratory Animals.

To generate LECT2 KO mice, single guide RNA (sgRNA) for clustered regularly interspaced short palindromic repeats (CRISPR)/CRISPR-associated (Cas)9 gene editing were designed on the website developed by Feng Zhang Lab (http://crispr.mit.edu/). The template for *in vitro* transcription of sgRNA was amplified using forward primer contain T7 promoter and sgRNA targeting LECT2 (Lect2 sgRNA: 5′-TAATACGACTCACTATAGGGAGCCATAGCTGTCACACG TCGTTTTAGAGCTAGAA-3′) and reverse primer (5′-AAAAAAGCACCGACT CGGTG-3′). Zygotes of C57BL/6J mouse strains were used for microinjection and

transplanted to pseudopregnant recipient mice. *In vitro* transcribed Cas9 RNA (100 ng μl$^{-1}$) and sgRNA (50 ng μl$^{-1}$) were injected into the zygotes. Oviduct transfers were performed on the second day. To analyse the genotype, genomic DNA was extracted with 50 mM sodium hydroxide from the LECT2 KO mice and PCR amplified using the forward and reverse primers (F: 5′-CATAGCCAGGGG ACTATGTTTTA-3′, R: 5′-ATATAGTCATAGCTGCACACAGCA-3′), with expected product sizes of 377 bp for the amplicon. PCR products were then submitted for Sanger sequencing.

**Humans.** This study was approved by the Ethics Committee of Ningbo University. Informed consent was obtained according to the Declaration of Helsinki. Human peripheral blood mononuclear cells were obtained from 31 healthy subjects (Ningbo No. 2 Hospital). The Ethics Committee of Ningbo No. 2 Hospital approved the use of discarded human blood (protocol #KYLL2013002). All subjects gave informed consent prior to blood collection.

**Reagents.** The recombinant LECT2 proteins were produced from Chinese hamster ovary (CHO) cells. The mouse LECT2 cDNA fragment was cloned into the EcoR I and Kpn I site of an expression vector pcDL-SRα296 provided by Biovector Science Lab to generate a LECT2 expression plasmid named pcDL-SRα296-mLECT2. CHO cells transfected with pcDL-SRα296-mLECT2 were maintained in Eagle's minimal essential medium supplemented with 10% fetal bovine serum. Recombinant LECT2 protein was purified from the culture fluids of CHO cells. Endotoxin in the recombinant proteins was less than 0.1 EU mg$^{-1}$ after toxin removal with an endotoxin-removal column (Pierce). LECT2 or G-CSF (Filgastrim, Amgen) was s.c. injected at a dose of 300 μg kg$^{-1}$ body weight (once a day for 5 days). The mice were killed 4 h after the final injection. For kinetic studies, the number of LSK cells in the blood were analysed at 1, 2, 4, 8, 12, 24 h, 2 d, 3 d, 4 d, or 5 d after LECT2 injection. 5FU (Sigma-Aldrich) was injected i.v. (150 mg kg$^{-1}$ body weight). In combination with LECT2 administration, anti-G-CSF IgGs and anti-G-CSFR IgGs (R&D Systems) were i.p. administered (300 μg kg$^{-1}$ body weight daily) for 5 consecutive days. For the G-CSF and LECT2 comparison, mice were exposed to 6 Gy (Cammacell-40, Atomic Energy of Canada Lim, Ottawa, Canada) 30, 25 or 20 days before the G-CSF or LECT2 treatment. A c-MET inhibitor (PHA-665752, Pfizer, 7.5 mg kg$^{-1}$, twice per day) and anti-CD209a IgG (GL Biochem, 200 μg kg$^{-1}$, daily) were delivered (i.p.) for five consecutive days in combination with LECT2. The CD169$^{DTR/+}$ mice were injected (i.v.) with diphtheria toxin (10 μg kg$^{-1}$ body weight) on the first and fourth days, and the mice were analysed on the seventh day. TNF (R&D Systems, Inc., 1.0 μg kg$^{-1}$ day$^{-1}$) was injected (i.v.) once per day combined with LECT2. To analyse osteolineage cell function, the mice were administered SrCl$_2$ (4 mmol kg$^{-1}$ per day, Sigma) via their drinking water for 12 weeks. For LECT2 neutralization, the mice received injections (i.p.) of a polyclonal anti-LECT2 IgG (100 μg per mouse per day, GL Biochem) for 3, 5 or 7 days. For the CXCR4 blockade experiment, AMD3100 (5 mg kg$^{-1}$, Sigma) was injected (s.c.) once at 1 h before killing on day 5. Diprotin A (5 μmol per mouse, Sigma) was injected (s.c.) twice per day on days 4 and 5, combined with an injection (s.c.) of LECT2 for five consecutive days. The N-terminal truncated mouse SDF-1 (SDF-1 (3–68)) protein was produced by treating mouse SDF-1 (R&D Systems) with CD26 (Sigma) for 18 h at 37 °C. The blood and BM were collected and analysed after an intra-femur injection (10 ng SDF-1 or SDF-1 (3–68) in 5 μl PBS). The macrophages were depleted using clodronate liposomes (0.2 ml per mouse i.v.)[12].

For the *in vitro* studies, the following reagents were used: TNF (10 ng ml$^{-1}$), etanercept (Enbrel, Pfizer, 10 nM).

**Flow cytometry.** The flushed BM cells and peripheral blood mononuclear cells were stained according to standard procedures, and the samples were analysed on a flow cytometer (Gallios, Beckman Coulter). The analysis was performed using Kaluza software (Beckman Coulter) and FlowJo software (Tree Star). The following fluorescently labelled antibodies were used: anti-mouse lineage cocktail (fluorescein isothiocyanate (FITC)-anti-CD4 (clone GK1.5, 1:200), FITC-anti-NK1.1 (clone PK136, 1:100), FITC-anti-CD11b (clone M1/70, 1:200), FITC-anti-B220 (clone RA3-6B2, 1:100), FITC-anti-Gr-1 (clone RB6-8C5, 1:200), FITC-anti-CD8a (clone 53-6.7, 1:100)), allophycocyanin (APC)-anti-c-Kit (clone 2B8, 1:50), phycoerythrin (PE)-anti-Sca-1 (clone D7, 1:100), FITC-anti-F4/80 (clone BM8, 1:400), peridinin-chlorophyll proteins-Cy5.5 (Percp-Cy5.5)-anti-CXCR4 (clone L276F12, 1:100), Pacific blue-anti-mouse CD45.1 (clone A20, 1:100), brilliant violet 570-anti-mouse CD45.2 (clone 104, 1:50), PE-Cy5-anti-mouse-Flk2 (clone A2F10, 1:50), PE-anti-human CD45 (clone HI30, 1:25), FITC-anti-Ter119 (clone TER-119, 1:100), FITC-anti-CD45 (clone 30-F11, 1:400), FITC-anti-CD31 (Clone 390, 1:100), and FITC-anti-Sca-1 (clone E13-161.7, 1:100) from BioLegend; anti-human lineage cocktail (APC-anti-human-CD2 (clone RPA-2.10), APC-anti-human-CD3 (clone OKT3), APC-anti-human-CD10 (clone SN5c), APC-anti-human-CD11b (clone CBRM1/5), APC-anti-human-CD14 (clone 61D3), APC-anti-human-CD16 (clone CB16), APC-anti-human-CD19 (clone HIB19), APC-anti-human-CD56 (clone MEM188), APC-anti-human-CD235a (clone HIR2), 1:5), Alexa Fluor 700-anti-mouse-CD34 (clone RAM34, 1:50), and PE-anti-CD51 (clone RMV-7, 1:100) from eBioscience; FITC-anti-human-CD34 (clone 581, 1:10), Phycoerythrin-Cy7 (PE-Cy7)-anti-human-CD38 (clone HIT2,

1:50), PE-anti-human-CD90 (clone 5E10, 1:100), phycoerythrin-Cy7 (PE-Cy7)-anti-ki-67 (clone B56, 1:50), biotin-anti-mouse CD209a (clone 5H10, 1:100), and streptavidin-PerCP (1:5) from BD Biosciences; and PE-anti-CD169 from R&D Systems (clone 645608, 1:20). For CD209a staining, biotin-anti-mouse CD209a was detected with streptavidin-PerCP. For intracellular Ki-67 and 7-amino actinomycin D (7-AAD, BioLegend) staining, the BM cells were surface-stained for LSK cell markers, fixed in fixation/permeabilization buffer (BD Biosciences), washed in Perm/Wash (BD Biosciences), and stained with PE-Cy7-anti-Ki-67. The cells were washed in Perm/Wash and resuspended in PBS/2% fetal calf serum (FCS, Invitrogen), and 7-AAD was then added to the samples.

**CFU assays.** The peripheral blood mononuclear cells were collected from the mice using standard techniques. We plated 10 μl of blood in 2.5 ml methylcellulose medium supplemented with a cocktail of recombinant cytokines (MethoCult 3434; STEM CELL Technologies). After 7 days in culture, the number of colonies per dish was counted.

**Immunofluorescence.** The femurs were isolated from the mice, post-fixed in 4% paraformaldehyde at 4 °C overnight, and decalcified in 10% EDTA at 4 °C for 1 week. Longitudinal sections of the femoral bones were prepared using a cryostat (CM1950, Leica). The sections were rinsed three times with PBS and blocked in 2% BSA/PBS for 30 min. The slides were incubated overnight at 4 °C with polyclonal anti-osteopontin (Santa Cruz Biotechnology) (1:50) and rinsed three times with PBS. The slides were incubated with PE-anti-goat-IgG (1:100) for 45 min at room temperature after the primary antibody incubation. The coverslips were mounted with SlowFade Gold Antifade reagent supplemented with the 4,6-diamino-2-phenylindole dihydrochloride (DAPI) nuclear stain (Invitrogen). Confocal images were captured using a Zeiss LSM 780 microscope (Carl Zeiss).

**Real-time quantitative PCR.** The CD209a$^+$ BM cells were sorted using the MoFlo XDP cell sorter (Beckman Coulter). The total RNA was extracted and purified from the steady-state or CD209a$^+$ BM cells using RNAiso reagents (TaKaRa). After deoxyribonuclease I treatment, the cDNAs were synthesized using reverse transcription M-MLV (TaKaRa). The following primers were used (forward and reverse, respectively): CD169, 5′-AGTGAGCCACCTGCTGAGAT-3′ and 5′-CCCAGTGTATTCTGGGCTGT-3′; RunX2, 5′-CCCAGCCACCTTTACCT ACA-3′ and 5′-TATGGAGTGCTGCTGGTCTG-3′; nestin, 5′-CTGCCTGGATCA CCCTGAAG-3′ and 5′-TACTGTAGACAGGCAGGGCT-3′; CTR, 5′-TAGTTAG TGCTCCTCGGGCT-3′ and 5′-AGTACTCTCCTCGCCTTCGT-3′; CD31, 5′-ATGACCCAGCAACATTCACA-3′ and 5′-TCGACAGGATGGAAATCACA-3′; myeloperoxidase (MPO), 5′-GACAACACTGGCATCACCAC-3′ and 5′-ATA GCACAGGAAGGCCAATG-3′; TNF, 5′-GCCTATGTCTCAGCCTCTTCTC-3′ and 5′-CACTTGGTGGTTTGCTACGA-3′; CCL3, 5′-CTGCCCTTGCTGTTCTT CTC-3′ and 5′-GTGGAATCTTCCGGCTGTAG-3′; CD209a, 5′-CACTGCCTGC CACAATGT-3′ and 5′-CCCAGTACCATGTAGACTCC-3′; G-CSF, 5′-GGAAG GAGATGGGTAAAT-3′ and 5′-GGAAGGGAGACCAGATGC-3′ and 18S rRNA, 5′-TTTGTTGGTTTTCGGAACTGA-3′ and 5′-CGTTTATGGTCGGAACTA CGA-3′. RT-qPCR was performed using SYBR premix Ex Taq (Perfect RealTime; TaKaRa). The data were normalized to the 18S rRNA.

**Cell isolation and culture.** The osteolineage cells were isolated[20]. The tibias, femurs and humeri of the mice were flushed thoroughly to remove the BM cells, chopped with a scalpel, and washed three times using a 5-ml polystyrene tube with a strainer (BD Biosciences) to further remove the residual BM cells. The bone fragments were then digested at 37 °C with Type IA collagenase (Sigma) for 1 h. The CD209a-positive osteolineage cells were detected using flow cytometry.

The parietal bones were obtained from newborn mice for the osteolineage cell cultures. After washing, the osteolineage cells were obtained by digestion with Type IA collagenase. For the macrophage cultures, the macrophages were isolated from BM cells using magnetic-activated cell sorting (MACS, Miltenyi Biotec). For the cytokine expression assay, the osteolineage cells or macrophages were incubated with the BM supernatant for 24 h, and the cell culture supernatants were removed. The cells were incubated with LECT2 (5 μg ml$^{-1}$) for another 3 days, and the supernatants were collected for further experiments.

**Transplantation experiments.** For the analysis of survival after transplantation, LSK cells were sorted using the MoFlo XDP cell sorter (Beckman Coulter) and Lin$^-$c-Kit$^+$ cells were MACS (Miltenyi Biotec) sorted from BM of mice that had been treated with PBS or LECT2 for 3 days. The LSK cells were injected (i.v.) into lethally irradiated (9 Gy) mice. In the sepsis model, $1 \times 10^5$ Lin$^-$c-Kit$^+$ cells were injected (i.v.) into *Pseudomonas aeruginosa*-infected mice.

For competitive bone marrow transplantation, the congenic Ly5.1/Ly5.2 system was employed[53]. LSK cells were sorted from the BM of the mice that had been treated with PBS or LECT2 for 3 days. The lethally irradiated B6.SJL (CD45.1) mice were transplanted with LSK cells from either the PBS- or LECT2-treated C57BL/6 mice (CD45.2) in competition with an equal amount of LSK cells from the SJL mice (CD45.1). Reconstitution of donor peripheral blood myeloid and lymphoid cells was monitored by staining with antibodies against CD45.2, B220 and CD11b.

For the second transplantation, CD45.2$^+$ LSK cells were sorted from BM isolated from the chimaeras of the first transplantation and transplanted as above.

For the analysis of osteolineage cell role, BM cells ($5 \times 10^6$) were obtained from CD209a KO mice. BM cells ($5 \times 10^6$) were injected (i.v.) into lethally irradiated (9 Gy) WT mice. The analysis of LSK cells in BM was performed 8 weeks after reconstitution.

For the mobilized blood cell transplantations, the lethally irradiated B6.SJL (CD45.1) mice were transplanted with 20 μl blood from six pooled mobilized C57BL/six mice (CD45.2) in competition with $2 \times 10^5$ BM cells from the SJL mice (CD45.1). Moreover, the lethally irradiated recipient B6.SJL mice (CD45.1) were also transplanted with $2 \times 10^5$ competitive whole BM cells from untreated B6.SJL mice (CD45.1) mixed with 5–80 μl blood from six pooled mobilized C57BL/six mice (CD45.2). Reconstitution of donor peripheral blood was further employed to estimate the RU and CRU values.

RU values were calculated according to Harrison's method[54]. Each RU represents the repopulating activity of $1 \times 10^5$ BM cells. This formula can be rearranged as: donor RU = % donor × $C$/(100 − % donor). $C$ is the number of competing BM RUs co-transplanted with the donor cells and $C = 2$ for $2 \times 10^5$ competitor BM cells. Limiting dilution analysis was performed[55]. After 20 weeks, the relative contribution of CD45.2 to total CD45 was established by flow cytometric analysis. Animals with >1% donor contribution were considered positive for donor cell engraftment. CRU frequency was calculated using L-Calc software (Stem Cell Technologies).

**NOD–SCID mice.** The NOD–SCID mice (NOD/LtSzPrKdcscid/PrKdcscid63) were sublethally irradiated with 3.75 Gy and injected (i.v.) with $2 \times 10^7$ human cord blood mononuclear cells 24 h later. HSC mobilization was performed 4–5 weeks after transplantation.

**Proliferation analysis.** To determine the proliferation status of the cells, mice were given two daily doses of BrdU (3.3 mg per mouse) by injection (i.p.) for three consecutive days, together with LECT2, and killed for analysis 3 h following the last administration.

**Enzyme-linked immunosorbent assay.** The levels of the LECT2, SDF-1 (total protein), CCL3, TNF, interleukin (IL)-6, IL-1β, CXC chemokine ligand 10 (CXCL10) and CCL4 proteins were analysed by enzyme-linked immunosorbent assay (ELISA). The LECT2 protein was detected by ELISA[12]. This ELISA system was based on one antibody as the capture antibody (rabbit anti-LECT2, C-terminal; Santa Cruz Biotechnology, Inc.) and another antibody for detection (goat anti-LECT2, N-terminal; Santa Cruz Biotechnology, Inc.). The remaining ELISA kits were obtained from R&D Systems. For ELISA to detect SDF-1(1–68), K15C mAb were obtained from Merck Millipore to recognize intact SDF-1 (ref. 56). The methods were conducted according to the manufacturer's instructions.

**Transcriptomic analysis.** After mice treated with LECT2 for 0, 3 and 5 days, the flushed BM cells were collected for transcriptomic analysis. The RNA-sequencing library was prepared using an Illumina TruSeq RNA sample prep kit (no. FC-122-1001). A total of 10 fmol of the library fragments was loaded into cBot to generate clusters, followed by sequencing on an Illumina HiSeq 2000. Gene expression was quantitated using Cufflinks 1.0.3. The data were deposited in the Gene Expression Omnibus (GEO) database under accession no. GSE70014.

**Enzyme activity analysis.** The CD26 peptidase activity (U/1,000 cells; 1 U = 1 pmol ρ-nitroanilide per minute) of the cells was measured in 96-well microplates using the chromogenic substrate Gly-Pro-p-nitroanilide (Gly-Pro-pNA, Sigma)[35].

**Statistical analyses.** The data represent the means ± s.e.m. The biological repeats are indicated by '$n$'. Sample size was chosen based on preliminary data and observed effect sizes. The mice used in the experiments were randomly chosen from our in-house colonies or suppliers. Experiments regarding immuno-fluorescence and animals were performed by an observer blinded to experimental conditions. We calculated the correlations between data sets using Pearson's correlation coefficient and the SPSS (version 13.0) software. We analysed the survival curves using the Kaplan–Meier method. The remaining data were analysed by one-way ANOVA. When variances were significantly different ($P < 0.05$), logarithmic transformation to stabilize the variance. If the normal distribution was not valid, statistical significance was evaluated using the Mann–Whitney $U$-test (two-tailed). *, ** and *** represent $P$ values <0.05, 0.01 and 0.001, respectively.

**Data availability.** RNA Sequencing data have been deposited in Gene Expression Omnibus (GEO) under accession number no. GSE70014. The authors declare that the remaining data are contained within the Article and Supplementary Information files or available from the corresponding author upon request.

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

## Acknowledgements

We thank Professor Tsvee Lapidot, Department of Immunology, Weizmann Institute of Science, Rehovot, Israel, for his expertise and critical reading of the manuscript. We thank Dr Masato Tanaka, Tokyo University of Pharmacy and Life Sciences Hachiouji, Tokyo; Dr Kenji Kohno, Research and Education Center for Genetic Information, Nara Institute of Science and Technology, Takayama, Ikoma, Nara, Japan and RIKEN BioResource Centre for providing CD169-DTR mice. This study was supported by the Research Project of Chinese Ministry of Education (213017A), the Program for the National Natural Science Foundation of China (31372555, 31472300), Zhejiang Provincial Natural Science Foundation of China (LZ13C190001), the LiDakSum Marine Biopharmaceutical Development Fund, and the KC Wong Magna Fund in Ningbo University.

## Author contributions

X.J.L., Q.C., Y.J.R., G.J.Y., C.H.L., N.Y.X., C.H.Y., H.Y.W., S.Z. and Y.H.S. did experiments and data analysis; C.H.Y., H.X.W. and S.Z. collected clinical samples; X.J.L. and J.C. designed *in vivo* and *ex vivo* experiments; X.J.L. and J.C. wrote the manuscript and J.C. directed the study.

## Additional information

**Competing financial interests:** The authors declare no competing financial interests.

