## [Peer Review File · Nature Communications]

Editorial Note: Parts of this peer review file have been redacted as we could not obtain permission to publish the reports of Reviewer #3 .

Reviewer #1 (Remarks to the Author)

In this study, the authors describe a novel pathway by which the LECT2 cytokine induces HSPC expansion in the bone marrow and mobilization. Specifically, they show that LECT2, acting through macrophages and osteolineage cells, represses TNF production, which, in turn, results in CD26 activation and CXCL12 cleavage. These are novel observations with important biological implications. The contrast with G-CSF induced HPSC mobilization is particularly interesting (and surprising).

Major concerns

The major concern of this study is readability. By this reviewer's count, there are 77 figure components (not including subpanels and supplementary material). Many of the experiments are not adequately outlined, leaving the reader guessing as to the experimental design. This reviewer suggests that the authors consider condensing their manuscript to focus on key experiments. For example, there is no need to show WBC counts in every figure. The CCL3 experiments are preliminary and could be eliminated. The mitomycin C experiments are difficult to interpret and should be eliminated. By reducing the number of figures, the authors can provide a more clear rationale and design for each experiment.

Figure 1n. The claim that HSC function is increased by LECT2 is not convincing. Figure 1m measures short-term repopulating activity-not HSCs. Figure 1n is presumably a competitive repopulation assay. To assess HSC function, need to show multilineage time-dependent engraftment. N = 3 is not sufficient.

Minor concerns

Figure 1b. Need to verify that the LECT2 preparation does not contain LPS

Figure 1g. The % of KLS cells in G0 appears low compared to published data. The authors may want to check their assay. The dot plots are too small to interpret.

Figure 2d. The LSK number in the BM is down in LECT2 KO mice but normal in CD209a KO mice. Why?

Figure 2g needs a legend

Figure 2i is nearly impossible to interpret with this tiny image. Could be eliminated.

Figure 3c. Is the difference in LSK in the blood significant between LECT2 vs. PBS treated CD169DTR mice?

Figure 4a. Again, the image is too small to interpret. Define the arrow heads. Show landmarks to orient the reader.

Figure 7a. Why not simply to AMD3100 alone. The anti-CXCR4 and anti-SDF1 are difficult to interpret and add little

Figure 8g. What dose of irradiation was used.

Figure 9b. How did you quantify osteolineage cells. This is difficult to do.

Reviewer #2 (Remarks to the Author)

This manuscript is interesting and the first to report that the chemokine LECT2 mobilizes hematopoietic stem cells (HSC) and progenitor cells (HPC) into blood. Generally well performed experiments show that daily administration of LECT2 protein in mice causes mobilization of colony-forming units and phenotypic HSPC (Lin- Kit+ Sca1+). However demonstration that long-term-reconstituting HSCs in transplantation setting are mobilized is lacking.

The mechanistic aspect of this work is well developed. Using blocking antibodies, inhibitors and KO mice, the authors convincingly demonstrate that the mobilizing effect of LECT2 is mediated by CD209a (or DC-SIGN) but not by its second receptor c-MET. Furthermore, LECT2 administration accelerates leukocyte recovery following cytotoxic challenge with 5-fluorouracil whereas CD209a gene deletion in KO mice compromises recovery. As CD209a+ sorted cells are found to express CD169 (macrophage restricted antigen) and RUNX2 (essential to osteolineage cells and their maturation) and the protein co-localized with CD169 and osteopontin, this suggests that the effect of LECT2 is indirectly mediated by macrophages and/or osteolineage cells. Using CD169-DTR mice and clodronate loaded liposomes to deplete macrophages in vivo and biglycan KO mice which have defective osteoblasts and SrCl2 to stimulate bone formation, the authors show that depletion of macrophages partially reduces the mobilizing effect of LECT2. As biglycan gene deletion partially reduces mobilizing effect of LECT2 and SrCl2 increases it, one may conclude that the effect is mediated by both macrophages and osteolineage cells. To further dissect the mechanisms, the authors show that LECT2 treatment reduces TNF expression and that administration of recombinant TNF abolishes the mobilizing effect of TNF whereas deletion of the TNF gene causes a small but significant leakage of HSPC into the blood and important loss of mobilization in response to LECT2. Together these data suggest that down-regulation of TNF secretion is an important step of HSPC mobilization in response to LECT2. As disruption of the chemotactic interaction between the chemokine SDF-1 and its receptor CXCR4 is known to play a major role in the mobilizing effect of many cytokines, the authors investigated this aspect by showing that LECT2 increases CD26 enzymatic activity in the marrow, reducing the amount of intact (active) SDF-1 in the marrow whereas injection of TNF reduces CD26 activity. These experiments support the model describe in figure 10. Finally the authors finish their manuscript by showing convincingly in G-CSF receptor KO mice and CD209a KO mice that the mechanisms leading to mobilization in response G-CSF and LECT2 are independent as GCSFR KO mice mobilize in response to LECT2 whereas CD209a KO mice mobilize in response to G-CSF.

Despite the novelty and originality, well performed experiments providing novel mechanisms that support the model, detailed Material and Methods section, etc. several issues must be addressed by the authors before publication:

- 1) The authors' claim in title, abstract results and particularly the discussion that LECT2 expands HSCs in the BM is the weakest. It is based on the finding of higher number of Lin-Kit+Sca1+Flt3-CD34- cells in the BM following LECT2 treatment. While this phenotype is OK to detect long-term reconstituting HSC in steady-state, this phenotype is not reliable when mice are stimulated by an exogenous agent. Sca1 and CD34 are two activation antigens in the mouse: Sca1 is up-regulated in

response to interferons (Essers MAG et al Nature 2009;450:904), while CD34 is upregulated on dividing HSCs in mice challenged with 5-FU or G-CSF (Sato T et al Blood 1999;94:548 and Ogawa M et al Ann NY Acad Sci 2001;938: 139). Therefore, the authors must confirm their claim with a functional competitive repopulation assay to quantify the number of repopulating units in the BM of mice LETC2 treated mice versus saline treated mice. In the absence of this functional assay, expansion of LT-HSC is not definitively established and this claim should be toned down.

2) The authors claim that in humans, number of circulating Lin CD34+CD38-CD90+ HSCs is correlated with plasma concentration of LECT2 (fig 1a). This should be substantiated by giving Pearson correlation coefficient and p value. Also the authors should clarify whether this is in steady-state.

3) In figure 1, the authors make several transplantation assays on sorted mobilized LSK cells. While this is fine to show that the engraftment potential of equivalent number of LSK cells is increased following LECT2 treatment, it does not quantify the number of reconstituting HSCs mobilized per ml of blood. It would be good to quantify this in a competitive transplantation assay in which 20uL whole mobilized blood is transplanted in competition with 200,000 congenic BM cells and compare blood content in competitive repopulating units in mice mobilized with LECT2 versus G-CSF.

4) Fig 1m, the survival curve should also be plotted as number of transplanted LSK cells on the X axis versus percentage of mice that did not engraft on the Y axis. This would enable to calculate by Poisson statistic the frequency of reconstituting HSCs within the LSK populations from the control and LECT2 treated mice and determine whether these differences are statistically different. This calculation can be performed using the L-calc software that can be downloaded from the Stem Cell Technologies website.

5) As a general comment, flow cytometry dot plots in Fig 1d,1e,1g, 1h and immunohistofluorescence images in Fig 2i and 4a are too small to see anything and be of any use. Their size must be increased.

6) Fig 2g is difficult to understand as it not explained what are the black bars and the white bars on the histogram.

7) In fig 9a,b the phenotype used to count macrophages and osteolineage cells is not described in legend, Result or Materials and Methods sections. Please specify.

8) In the Results section, too many times irradiation doses, origin of the cells (BM or blood), etc are not specified in text or the corresponding figure legends. This forces the reader to go back and forth between the Materials Methods section and Results sections too many times which makes the reading tedious. In some instance, this information is not even given in the MM section. For instance what were the dose, schedule and route of Diprotin A administration in Fig 7?

9) In page 6, line 130, osteoblasts are not ablated in biglycan KO mice, otherwise they would have no bones. This sentence must be changed. These mice have slightly reduced numbers of osteoblasts however they are defective as bone formation and mineralization are more reduced resulting in an osteoporotic phenotype (Xu T et al, Nat Genet 1998;20:78).

Reviewer #1 (Expert in HSc mobilisation)

Major concerns

The major concern of this study is readability. By this reviewer's count, there are 77 figure components (not including subpanels and supplementary material). Many of the experiments are not adequately outlined, leaving the reader guessing as to the experimental design. This reviewer suggests that the authors consider condensing their manuscript to focus on key experiments. For example, there is no need to show WBC counts in every figure. The CCL3 experiments are preliminary and could be eliminated. The mitomycin C experiments are difficult to interpret and should be eliminated. By reducing the number of figures, the authors can provide a more clear rationale and design for each experiment.

Response: We thank the reviewer for this comment which helps us to improve our study. We removed the WBC data in Fig. 2 (Now Fig. 3), Fig. 6, Fig. 7. The data of CCL3 expression after LECT2 treatment were removed in previous Fig. 5 (now supplementary Fig. 7b,c.). Mitomycin C experiments were removed in previous Fig. 5 (now supplementary Fig. 7).

Figure 1n. The claim that HSC function is increased by LECT2 is not convincing. Figure 1m measures short-term repopulating activity-not HSCs. Figure 1n is presumably a competitive repopulation assay. To assess HSC function, need to show multilineage time-dependent engraftment. N = 3 is not sufficient.

Response: We thank the reviewer for his most helpful review and comments. We showed the multilineage time-dependent engraftment in Fig. 2d. The data were existed in this paper last year when submitted to other Journal. We removed them when the paper was submitted to Nature Communications this year. We also added new data in previous Fig. 1n (now Fig. 2c, $n = 5$). We also added the description of these data in Results in line 75-97, and the protocols in Methods in 535-537.

Minor concerns

Figure 1b. Need to verify that the LECT2 preparation does not contain LPS

Response: This is a very good point. Endotoxin in the recombinant proteins was less than 0.1 EU/mg after toxin removal with an endotoxin-removal column (Pierce). We also added the description in line 403. We also found that LECT2 treatment also enhanced the CFU-Cs, WBCs and LSK cells in the blood of C3H/HeJ mice, a strain that is relatively insensitive to endotoxin. We also added the experiment data in supplementary Fig. 1 and description in Results in line 63-65.

Figure 1g. The % of KLS cells in G0 appears low compared to published data. The authors may want to check their assay. The dot plots are too small to interpret.

Response: We agree with the reviewer's comment. Indeed, our data is low compared with published data (about 90% of published data). We repeated this experiment, and found that the average percentage of LSK cells in G0 was 39.4% in WT mice. We also put the bigger dot

plots in Fig. 1g. It is similar with published data, 39% (Blood, 2012, 120: 1843-1855), 38.7% (Blood, 2013, 121: 5158-5166), about 45% (Immunity, 2015, 42:1021-1032).

Figure 2d. The LSK number in the BM is down in LECT2 KO mice but normal in CD209a KO mice. Why?

Response: We agree with the reviewer's comment. The LSK numbers have about 10% fluctuation performed at same conditions in different time according to our experimental experience. We repeated the experiment with WT control in Fig. 2d (Now Fig. 3d). The LSK numbers in the BM of CD209a KO mice are about 70.8% of WT mice.

Figure 2g needs a legend

Response: We agree with the reviewer's comment and have added a legend in Fig. 2g (now Fig. 3g). The mRNA levels of BM cell markers in CD209a⁺ BM cells relative to the level in steady-state BM cells. CD169, a marker of macrophages; RunX2, a marker of osteolineage cells; nestin, a marker of mesenchymal stem or stromal cells; calcitonin receptor (CTR), a marker of osteoclasts; CD31, a marker of endothelial cells; myeloperoxidase (MPO), a marker of neutrophils.

Figure 2i is nearly impossible to interpret with this tiny image. Could be eliminated.

Response: We agree with the reviewer's comment. The data have been deleted.

Figure 3c. Is the difference in LSK in the blood significant between LECT2 vs. PBS treated CD169DTR mice?

Response: Yes, they have. We have marked it (now Fig. 4c) and increased the description in line 139.

Figure 4a. Again, the image is too small to interpret. Define the arrow heads. Show landmarks to orient the reader.

Response: We agree with the reviewer's comment. We have added the information of arrow heads and landmarks in Fig. 4a (Now Fig. 5a). Anatomical landmarks are indicated: bone marrow (BM) and trabecular bone (TB). The OPN positive cells are indicated by arrow heads.

Figure 7a. Why not simply to AMD3100 alone. The anti-CXCR4 and anti-SDF1 are difficult to interpret and add little

Response: This is a very good point. We removed the data of anti-CXCR4 and anti-SDF1 in Fig. 7.

Figure 8g. What dose of irradiation was used.

Response: We have added in line 222 and Fig. 8g. 6 Gy.

Figure 9b. How did you quantify osteolineage cells. This is difficult to do.

Response: We agree with the reviewer's comment. We have added the description of marker for osteolineage cells in Fig. 9b. The osteolineage cells were CD45-Ter119-CD31-Sca-1-CD51⁺ cells isolated from BM as previously described (Chow A,

Reviewer #2 (Expert in HSC, macrophage)

(Remarks to the Author):

1) The authors' claim in title, abstract results and particularly the discussion that LECT2 expands HSCs in the BM is the weakest. It is based on the finding of higher number of Lin-Kit+Sca1+Flt3-CD34- cells in the BM following LECT2 treatment. While this phenotype is OK to detect long-term reconstituting HSC in steady-state, this phenotype is not reliable when mice are stimulated by an exogenous agent. Sca1 and CD34 are two activation antigens in the mouse: Sca1 is up-regulated in response to interferons (Essers MAG et al Nature 2009;450:904), while CD34 is upregulated on dividing HSCs in mice challenged with 5-FU or G-CSF (Sato T et al Blood 1999;94:548 and Ogawa M et al Ann NY Acad Sci 2001:938: 139). Therefore, the authors must confirm their claim with a functional competitive repopulation assay to quantify the number of repopulating units in the BM of mice LECT2 treated mice versus saline treated mice. In the absence of this functional assay, expansion of LT-HSC is not definitively established and this claim should be toned down.

Response: We thank the reviewer for this comment which helps us to improve our study. We indeed added more description about HSC expansion in results (in line 75-97) and discussion (in line 265-280). The HSC expansion is mentioned in title and abstract. The word limitation prevents the further description of HSC expansion in title and abstract.

We showed the number of repopulating units in the 1×10^3 LSK cells from the BM of LECT2 treated and PBS treated mice in Fig. 2e. We also showed the number of repopulating units in the 1×10^5 BM cells from the BM of LECT2 treated and PBS treated mice in Fig. 2e. The data were existed in this paper last year when submitted to other Journal. We removed them when the paper was submitted to Nature Communications this year. We also cited the papers (Essers MAG et al Nature 2009;450:904; Sato T et al Blood 1999;94:548 and Ogawa M et al Ann NY Acad Sci 2001:938: 139) to suggest the reason for competitive repopulation assay in line 76.

2) The authors claim that in humans, number of circulating Lin CD34+CD38-CD90+ HSCs is correlated with plasma concentration of LECT2 (fig 1a). This should be substantiated by giving Pearson correlation coefficient and p value. Also the authors should clarify whether this is in steady-state.

Response: We agree with the reviewer's comment. We added the Pearson correlation coefficient, p value, and in steady state in line 57-59.

3) In figure 1, the authors make several transplantation assays on sorted mobilized LSK cells. While this is fine to show that the engraftment potential of equivalent number of LSK cells is increased following LECT2 treatment, it does not quantify the number of reconstituting HSCs mobilized per ml of blood. It would be good to quantify this in a competitive transplantation assay in which 20uL whole mobilized blood is transplanted in competition with 200,000 congenic BM cells and compare blood content in competitive repopulating units in mice mobilized with LECT2 versus G-CSF.

Response: We thank the reviewer for his most helpful review and comments. It is important to illustrate the competitive repopulating units (CRUs) and repopulating units (RUs) in LECT2 or G-CSF mobilized blood. We began to perform this experiment in last year and get the data now. We added this data in supplementary Fig. 9 and description in Results in line 215-218.

4) Fig 1m, the survival curve should also be plotted as number of transplanted LSK cells on the X axis versus percentage of mice that did not engraft on the Y axis. This would enable to calculate by Poisson statistic the frequency of reconstituting HSCs within the LSK populations from the control and LECT2 treated mice and determine whether these differences are statistically different. This calculation can be performed using the L-calc software that can be downloaded from the Stem Cell Technologies website.

Response: We thank the reviewer for his most helpful review and comments. We added CRU data by Poisson statistic the frequency of reconstituting HSCs within the LSK populations from the control and LECT2 treated mice in Figure 2b.

5) As a general comment, flow cytometry dot plots in Fig 1d,1e,1g, 1h and immunohistofluorescence images in Fig 2i and 4a are too small to see anything and be of any use. Their size must be increased.

Response: We thank the reviewer for this comment which helps us to improve our study. We zoomed up the Fig. 1d, 1e, 1g, 1i, and Fig. 4a (now Fig. 5a). Fig. 2i has been deleted according to the reviewer 1.

6) Fig 2g is difficult to understand as it not explained what are the black bars and the white bars on the histogram.

Response: We agree with the reviewer's comment. We have added the explanation of the black bars and white bars in Fig. 2g (now Fig. 3g).

7) In fig 9a,b the phenotype used to count macrophages and osteolineage cells is not described in legend, Result or Materials and Methods sections. Please specify.

Response: We agree with the reviewer's comment. We have added the phenotype of macrophages and osteolineage cells in legend in Fig. 9a,b and in results in line 240-241.

8) In the Results section, too many times irradiation doses, origin of the cells (BM or blood), etc are not specified in text or the corresponding figure legends. This forces the reader to go back and forth between the Materials Methods section and Results sections too many times which makes the reading tedious. In some instance, this information is not even given in the MM section. For instance what were the dose, schedule and route of Diprotin A administration in Fig 7?

Response: We agree with the reviewer's comment. We have added irradiation doses in line 153 and 222, and added more engraftment protocols in Figure 2. We added the dose, schedule and route of diprotin A administration in Results in line 199. We added the source of LSK cells in every transplantation experiment in line 75-97, line 215-218.

9) In page 6, line 130, osteoblasts are not ablated in biglycan KO mice, otherwise they would have no bones. This sentence must be changed. These mice have slightly reduced numbers of osteoblasts however they are defective as bone formation and mineralization are more reduced resulting in an osteoporotic phenotype (Xu T et al, Nat Genet 1998;20:78).

Response: We agree with the reviewer's comment. "depleted" has been changed to "reduced" in line 146.

Reviewer #1 (Remarks to the Author)

The authors have improved the readability of the manuscript significantly, and they have adequately addressed this reviewer's concerns.

Reviewer #2 (Remarks to the Author)

This revised manuscript is much improved and most of my comments have been addressed adequately. The authors have addressed my concerns in respect to the scientific content and interpretation of the results as performing functional transplantation assays with bone marrow and mobilized blood from their LETC2 treated mice. I only have minor corrections to suggest as several aspects of the presentation and the figures should be improved to ease the reading of the manuscript.

In response to my comment#1, transplantation experiments with bone marrow cells from LECT2 treated mice in Figs 2a and 2b support phenotypic results in Fig 1e as well as the authors' claim that LETC2 expands the HSC compartment in the bone marrow. Figure 2a could be improved by plotting % of surviving recipients versus number of transplanted LKS cells in order to calculate frequency of reconstituting cells by Poisson distribution and p values associated with these frequencies (similar to Fig 2b).

In response to my comment#3, transplantation of mobilized blood convincingly shows that blood mobilized with LETC2 contains more competitive repopulating cells in supplementary figure 9 which could be added directly to figure 9.

There is however still some editing to do in respect to the figures and some mistakes in the text. These should be corrected. I list the most apparent below.

1. Panels in all the figures of the main article are still far too small to see the details. In fact, the supplementary figures are much better (larger size and better resolution). Please make panels with similar size and resolution as in supplementary figures. For instance, all histograms (bar charts) have minuscule dots on the side, which seem to represent the result for each individual mouse for each group. This is important information but I struggled to see those and this is not explained in the legends. All flow cytometry dot-plots and histograms are far too small. It's particularly bad for Fig 1g (I can hardly see the dots in Go and S/G2/M gates, Fig 1i (can't see dots in CD34- gate), and Fig 3h. In fig 4a,d and Fig 7c the color scheme for the different profiles make some of them invisible, and immunohistofluorescence in Fig5a is far too small to see any detail.

2. In line 35, the word "extra-marrow" could be replaced by "systemic" or "extra-medullar".

3. In line 111, the word "down-regulated" should be replaced by "reduced". (numbers can't be down-regulated).

4. In line 147, the sentence "The number of LSK cells in the BM and blood decreased in osteolineage cell-depleted mice..." should be replaced by "The number of LSK cells in the BM and blood decreased in Bgn^{-/-} mice..." (see my comment#9 in my first review).

5. In line 343, the authors may wish to cite To LB et al Blood 2011;118:4530 in which the possible causes of failed HSC mobilization are reviewed in detail.
6. In line 349, both your references 49 (Christopher et al 2011) and 41 (Winkler et al 2010) show that BM resident macrophages are reduced in response to G-CSF. Both should be cited.
7. In line 353, I would be a bit more cautious and write "LETC2 may represent a potential HSC-mobilizing" agent.